# CURIOSITY IS THE PATH TO OPTIMIZATION

## ABSTRACT

In PAC(Probably Approximately Correct) theory, it is posited that larger hypothesis spaces necessitate more independently and identically distributed (i.i.d) data to maintain the accuracy of model performance. PAC-MDP theory defines curiosity by assigning higher rewards for visiting states that are far from the previously visited trajectory, which supports more independent and i.i.d data collection. Recently, this field has witnessed attempts to narrow the hypothesis space by developing additional mechanisms that train multiple skills and facilitate the sharing of information among them, thereby discovering commonalities. However, one might wonder: What if curiosity could not only enhance the efficiency of data collection but also significantly reduce the hypothesis space, thereby driving optimal outcomes independently without additional mechanism used in PAC-MDP? Within this context, contrastive multi-skill reinforcement learning (RL) exhibits both traits. Previous research in contrastive multi-skill RL has utilized this technique primarily as a form of pretraining, However, there has been scant investigation into whether the technique itself can reduce the hypothesis space to optimize the outcomes. We have mathematically proven that curiosity provides bounds to guarantee optimality in contrastive multi-skill reinforcement learning (RL). Additionally, we have leveraged these findings to develop an algorithm that is applicable in real-world scenarios, which has been demonstrated to surpass other prominent algorithms. Furthermore, our experiments have shown that different skills are actually reducing the hypothesis space of the policy by being hierarchically grouped.

## 1 INTRODUCTION

We have inherently evolved to possess curiosity, driving us to seek out and learn from areas of high uncertainty. This natural inclination enhances our intelligence by continually pushing the boundaries of our knowledge. In the realm of machine learning, this trait is mirrored in methodologies like Active Learning and the PAC-MDP (Probably Approximately Correct - Markov Decision Process), which strategically navigate uncertainty to boost learning efficiency. This focus not only accelerates learning processes but also ensures that model target the most valuable information, optimizing intelligence enhancement across diverse environments. (Auer et al., 2002; Raj & Bach, 2022).

We propose that humans instinctively adhere to general biases, consistent with the principle of minimal description length (Grünwald, 2007), and we underscore the efficacy of modularizing commonalities. Similarly, the PAC-MDP framework enhances reinforcement learning by offering theoretical guarantees, such as upper bounds of the necessary sample size to approximate the optimal policy. These bounds are crucial for minimizing the **hypothesis space**—the set of all possible policies an algorithm might consider—thereby ensuring more precise and predictable outcomes with reduced sample complexity.

However, unlike deliberate efforts to reduce the hypothesis space, we aim to discuss how curiosity itself serves as a process of reducing the hypothesis space. In this paper, we propose a novel reinterpretation of curiosity and, building upon this new understanding, we adapt the information gain formula typically employed in contrastive multi-skill reinforcement learning. This adaptation contributes to the autonomous optimization of the hypothesis space. This research demonstrates how skills self-categorize and inherently structure themselves into tree-like formations, effectively minimizing the hypothesis space. Finally, we present a practical modification of our algorithm, which we show surpasses the performance of existing algorithms.

Our contributions are summarized as follows:

- We have established a bound that guarantees optimality purely from curiosity in an unsupervised learning environment.
- We have adapted unsupervised learning algorithms, previously limited to finite state MDPs with optimality guarantees, to create algorithms that operate effectively in continuous state MDPs.
- Designing a contrastive space where skill itself serves as a dimension, allowing for non-parametric embedding among them.
- Revealing that skills, grouped hierarchically by shared traits, form coherent concepts.

## 2 RELATED WORK

**Accelerating PAC-MDP with Hypothesis Space Reduction** Previous research has successfully leveraged the observation that different policies trained to solve various tasks share common features. This method is akin to meta-reinforcement learning (meta-RL), where the available space for each policy is effectively reduced. In particular, it is assumed that different MDPs share beneficial options, a concept explored within the framework of Semi-Markov Decision Processes (SMDP) (Brunskill & Li, 2014). Further research has focused on grouping tasks that share the same MDP (Brunskill & Li, 2013), utilizing the best initialization from previous experiences to reshape rewards and accelerate task performance (Chu et al., 2021; Abel et al., 2018), and concurrently training multiple agents to solve their respective tasks while sharing information to accelerate each other's learning (Liu et al., 2016; Zhang & Wang, 2021; Sun et al., 2020; Dimakopoulou & Van Roy, 2018; Guo & Brunskill, 2015).

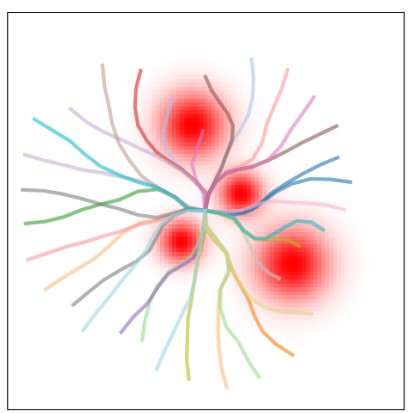

Figure 1: The result of training 32 skills contrastively using the algorithm we proposed shows that by making the policy itself a dimension and allowing the policies to embed each other non-parametrically, the skills autonomously organize into hierarchical groups. This organization is based solely on information gain, without the need for external rewards, and leads to an optimality guarantee within a specified confidence bound.

**Unsupervised Skill Discovery** The problem mode collapse is the issue where the vastness of the search space leads to insufficient exploration (Lee et al., 2020). Numerous efforts have been made to address this issue. For example, some RL focuses on states where the results differ from our predictions (Berseth et al., 2020; Pathak et al., 2017; 2019; Burda et al., 2019). However, this approach can cause the model to oscillate and become unstable, akin to the "noisy TV" problem. To address such complex challenges, several methodologies have been developed, including Hierarchical Reinforcement Learning (Levy et al., 2019; Vezhnevets et al., 2020), which strategically divides tasks into manageable subtasks. Furthermore, several approaches have been developed to train multiple skills, ensuring that when a new skill is autonomously acquired, it remains clearly distinct from previously learned skills. VIC (Gregor & Danilo Jimenez Rezende, 2016) involves training multiple policies simultaneously and subsequently ensuring that they diverge from each other. However, because it learned both the distribution of skills and the distribution of states simultaneously, the mode collapse problem still occurred. Subsequently, DIAYN (Eysenbach et al., 2019)and VALOR (Achiam et al., 2018) sought better exploration by fixing the distribution of skills and adding an entropy term to the skills. To avoid mode collapse and achieve better exploration, some approaches modified mutual information formula from backward to forward, artificially rewarding the model to spread the state distribution as widely as possible during learning (Lee et al., 2020; Campos et al., 2020; Sharma et al., 2020). APT (Liu & Abbeel, 2021b), APS (Liu & Abbeel, 2021a) and CIC (Laskin et al., 2022) aimed to find more optimized patterns not by increasing state coverage but by simultaneously learning state embeddings during the feature training process and maximizing the distribution of the embedded space. In APS (Liu & Abbeel, 2021a) and CIC (Laskin

et al., 2022), they maximized the state entropy by separately defining the discriminator term and the exploration term, making them come from different mechanism. Recently, algorithms have been developed that offer improved performance in environments where the optimal path to a given state is provided METRA (Seohong Park & Levine, 2024).

**Optimality in Contrastive learning** The following two studies are most closely related to our research, as our mechanism addresses optimality with respect to any reward function. In the study referenced in (Eysenbach et al., 2022), in a finite-state MDP environment, researchers demonstrated that maximizing information gain in skills yields optimal results for some reward function by allocating the skill distribution in its vertex of feasible set. Similarly, the paper (Mutti et al., 2022) demonstrated that by using Rényi divergence in a finite state MDP to create maximally diverse skills, ensuring that the distance between any skill and a set of specific skills remains below a defined threshold, thereby cumulatively enabling the establishment of a performance bound between the optimal policy and the learned policies. This study locates skills at the vertices of the feasible set within a continuous state, continuous action environment. Simultaneously, it establishes a boundary condition ensuring that the distance between any given skill and those skills developed through our learning process remains below a certain threshold.

# 3 PRELIMINARIES

## 3.1 CONTRASTIVE SPACE

Consider the spaces $\mathcal{S} = \mathbb{R}^n$ and $\mathcal{Z} = \mathbb{R}^z$. Let $(\mathcal{S}, d)$ denote a metric space where the distance $d$ between any two points $x_1, x_2 \in \mathcal{S}$ is defined as follows:

$$d(x_1, x_2) = \|x_1 - x_2\|$$

Define a projection function $\psi : \mathcal{S} \to \mathcal{Z}$, where $\mathcal{Z}$ represents the contrastive space. The contrastive space is conceptualized such that the dimensions can embody policies.

The distance between two projected points $y_1$ and $y_2$ in $\mathcal{Z}$, corresponding to the original points $x_1$ and $x_2$, is given by the Euclidean distance in $\mathbb{R}^z$:

$$d_{\mathcal{Z}}(y_1, y_2) = \|\psi(x_1) - \psi(x_2)\|$$

where $y_1 = \psi(x_1)$ and $y_2 = \psi(x_2)$.

Let $Z$ be the set of skills and $T$ the set of trajectories. Let $B$ denote the dimensional bandwidth, which serves as an indicator of confidence, representing how confidently a skill is positioned within a certain state. We define $\psi$ as follows, where $i$ and $j$ are elements of $Z$:

$$\psi_B(s_{i,t})_j = \begin{cases} \frac{1}{T} \sum_{t'=1}^{T} \exp\left(-\frac{d(s_{j,t'}, s_{i,t})^2}{B}\right) & \text{if } i \neq j, \\ 1 & \text{if } i = j. \end{cases}$$

## 3.2 PERFORMANCE MEASUREMENT IN MARKOV DECISION PROCESSES

In this study, we conduct experiments across a variety of MDP environments, each characterized by a unique set of reward functions. To accommodate the complexity of these environments, we employ a Multi-Objective Markov Decision Process (MOMDP), formally defined as tuple $\mathcal{M} := \langle \mathcal{S}, \mathcal{A}, \mathcal{P}, \mathcal{R}, \gamma \rangle$. Unlike a standard MDP, where $R$ represents the reward function, in a MOMDP, $R$ consists of multiple reward functions corresponding to different objectives. The elements of the MOMDP are mathematically described as follows:

- **States** $\mathcal{S}$: An infinite and bounded set of states, where $s \in \mathcal{S}$ represents an individual state from the set of states.
- **Actions** $\mathcal{A}$: An infinite and bounded set of actions available to the agent, defining possible movements that can be taken in each state. where $a \in \mathcal{A}$ represents an individual action

from the set of actions. drawn as $a_t \sim \pi_\theta(a_t|s_t; z) + \epsilon$ at each time step t with learnable policy $\pi$ with parameter $\theta$, with Gaussian noise $\epsilon$ influencing the outcome.

- **Transition Probabilities** $\mathcal{P}$: The transition model $\mathcal{P} : \mathcal{S} \times \mathcal{A} \to \Delta(\mathcal{S})$ defines the system dynamics, where the subsequent state $s$ is probabilistically determined according to the distribution $\mathcal{P}(\cdot \mid s, a)$ given the current state $s$ and action $a$.

- **Discount Factor** $\gamma$: A discount factor, $\gamma \in [0, 1)$, used to decrease the value of future rewards, reflecting the preference for immediate rewards over distant ones.

- **Set of the Reward Functions** $\mathcal{R}$: We define set of state dependent objective function $\mathcal{O} = \{\mathbf{O} \mid \mathbf{O} : \mathcal{S} \to \mathbb{R}\}$, and set of the reward function $\mathcal{R} = \{\mathbf{R} \mid \mathbf{R} : \mathcal{S} \times \mathcal{S} \to \mathbb{R}\}$, where $\mathbf{R} = (R_1, R_2, \dots, R_{|Z|})$ and each $R_z : \mathcal{S} \times \mathcal{S} \to [-R_{\max}, R_{\max}]$. Here, $R_z(s_{z,t}, s_{z,t+1}) = O_z(s_{z,t+1}) - O_z(s_{z,t})$, and $R_{\max}$ represents the maximum value that any reward function can take.

Given that the space contains infinitely many states, we modeled the state distribution $\mu$ induced by policy $\pi_{z,\theta}$ as a Gaussian Mixture Distribution. Instead of determining the state distribution based on the visitation frequency for each state, we assumed a Gaussian distribution centered at each state to estimate the state distribution effectively. It can be expressed as:

$$\mu_{z,\theta}(x) = \frac{1}{T} \sum_{t=0}^{T} \mathcal{N}(x|s_{z,t}, C \cdot E).$$

In this context, C denotes a variance small enough that the distribution closely approximates a point distribution. where $E$ is the identity matrix. The covariance matrix is then given by $C \cdot E$. In this configuration, each mean $s_t$ is located at a point along the trajectory generated by the policy.

Without loss of generity, we put $O_z(s_{z,0})$ as 0. We define the performance measure $J_\theta(z)$, where the discounted reward $R'(s_{z,t}, s_{z,t+1})$ at time $t$ is given by $\gamma^t R(s_{z,t}, s_{z,t+1})$, as follows:

$$J_\theta(z) = \mathbb{E}^\pi \left[ \sum_{t=1}^{T} \gamma^t R_z(s_{z,t}, s_{z,t+1}) \right] \tag{1}$$

$$= \mathbb{E}^\pi \left[ \sum_{t=1}^{T} \gamma^t O_z(s_{z,t+1}) - \gamma^t O_z(s_{z,t}) \right] \tag{2}$$

$$= \frac{1}{\gamma} \mathbb{E}^\pi \left[ \sum_{t=1}^{T} \gamma^t O_z(s_{z,t}) \right] - \mathbb{E}^\pi \left[ \sum_{t=0}^{T-1} \gamma^t O_z(s_{z,t}) \right] \tag{3}$$

$$= \frac{1-\gamma}{\gamma} \mathbb{E}^\pi \left[ \sum_{t=1}^{T-1} \gamma^t O_z(s_{z,t}) + \frac{\gamma^T}{1-\gamma} O_z(s_{z,T}) \right] \tag{4}$$

$$= \mathbb{E}^\pi \left[ \sum_{t=1}^{T} \gamma(t) O_z(s_{z,t}) \right] = \mathbb{E}^\pi \left[ \sum_{t=1}^{T} O'_z(s_{z,t}) \right] \tag{5}$$

Since $O'$ is derived by applying a discount factor to $O$, it respects $T R_{\max}$ as an upper bound.

## 4 CONCEPT BLOCK

### 4.1 COVERING OPTIMAL POLICY SPACE WITH MINIMAL HYPOTHESIS SPACE

We initially define the objective function set $\mathcal{O}^*$ and the set of optimal policies, denoted by $Z^* = \{\pi_{1,\theta^*}, \pi_{2,\theta^*}, \dots, \pi_{|Z|,\theta^*}\}$, where $\theta^*$ specifies the parameters that optimize each policy within the set. such that, regardless of the specific optimal policy introduced, The performance difference with any given policy is bounded by a certain threshold, with the objective of constructing a minimal hypothesis space $\theta^* \in \Theta$.

**Definition 4.1.** *(Separating objective function set) Define a separating objective function set $\mathcal{O}^*$ consisting of $|Z|$ objective functions $O_z^*$, each tailored to a specific policy $z$ in $Z$. These reward functions are selected to maximize the minimum performance differences between each policy $z \in Z$ and all other policies $z' \in Z$, $z' \neq z$. More precisely,*

$$O_z^* = \arg\max \min_{z' \in Z, z' \neq z} |\mathbb{E}_{x \in S}[\mu_{z,\theta}(x)O_z'(x)] - \mathbb{E}_{x \in S}[\mu_{z',\theta}(x)O_z'(x)]|.$$

**Definition 4.2.** *(Optimal policy set) For each skill $z$ and for each separating objective function set $O_z^*$, denote the minimum performance difference achieved as $\zeta_z$, given by:*

$$\zeta_z = \min_{z' \in Z, z' \neq z} |\mathbb{E}_{x \in S}[\mu_{z,\theta}(x)O_z^*(x)] - \mathbb{E}_{x \in S}[\mu_{z',\theta}(x)O_z^*(x)]|.$$

*Consider the set of these minimum performance differences:*

$$\zeta_Z = \{\zeta_z : z \in Z\}.$$

*The concept of a optimal skill set $Z^*$ generated by $\pi_{\theta^*}$ is having a maximized set of performance differences $\zeta_{Z^*}$.*

$$\theta^* = \arg\max_{\theta \in \Theta} \mathbb{E}[\zeta_z(\theta) : z \in Z].$$

The optimal policy derived in this manner possesses the following density property:

**Theorem 4.1.** *(Density property) The value of $\zeta_{Z^*}$ for an optimal policy set $Z^*$ does not increase upon adding another optimal policy $z_{|Z|+1}$ to the set $Z^*$.*

$$|J_\theta(z) - J_{\theta^*}(z)| \leq |J_\theta(z) - J_\theta(z')|$$

As seen in Equation (1) of Chapter 3 in the work by (Mutti et al., 2022), the problem of finding the optimal policy set as defined above is equivalent to the set cover problem and is known to be NP-hard (Feige, 1998). In the next chapter, we will introduce strategic approaches for asymptotically solving this problem.

### 4.2 CONVEX OPTIMIZATION AND INFORMATION GAIN

As we mentioned previously, maximizing the performance between every pair of policies, represented by $\zeta_{Z^*}$, is a challenging endeavor. Since each skill always attains the maximum value along the axis it represents in contrastive space, we first maximize their distance from the mean in the contrastive space as an approach to approximating a solution to this problem:

$$\max \sum_{i \in S_{z_i}} \left| \psi_B(x_i) - \frac{1}{|S_{z_j}|} \sum_{j \in S_{z_j}} \psi_B(x_j) \right| \tag{6}$$

However, directly maximizing the difference in distributions does not always place policies on the vertex of the policy space's convex set, but they also assign them within the interior of the set as shown in Figure 3 (left).

Consider the following scenario: Suppose there are four distinct skills. We analyze their positioning within a contrastive space in two different cases:

- **Case 1:**
    - Skills are positioned as: $(1, 0, 0, 0)$, $(0, 1, 0, 0)$, $(0, 0, 1, 0)$, and $(0, 0, 0, 1)$.
    - The minimum distance between any two skills is 2.
    - The average distance across all pairs is also 2.
- **Case 2:**
    - Skills are positioned as: $(1, 0.1, 0.1, 1)$, $(0.1, 1, 1, 0.1)$, $(1, 0.1, 1, 0.1)$, $(0.1, 1, 0.1, 1)$.
    - The shortest distance between two skills is reduced to 1.8.
    - The average distance between pairs increases to 3.

As can be seen in this case, efforts to increase the average distance across all pairs may hinder efforts to increase the minimum distance between any two skills.

To avoid this problem, we utilized Principle Lemma 6.1 from the MISL (Eysenbach et al., 2022) to optimize the allocation of policies within the vertices of the policy space. We can interpret each of the normalized $|Z|$ dimensions in the contrastive space as state occupancy measures of $|Z|$ state MDPs. Additionally, the reward function in each finite state MDP can be understood as a linear combination of the axes in $|Z|$-dimensional space.

According to Lemma 6.1 of (Eysenbach et al., 2022), when the information gain among policies is maximized, the policies are strategically positioned at the vertices of a feasible set within the contrastive space, as described in Figure 2. In our analysis, we calculate the information gain among policies assuming that each policy follows a Gaussian distribution. We further assume that the variance of each distribution is identical and denoted by $\delta$, and is sufficiently small to be considered as point distributions.

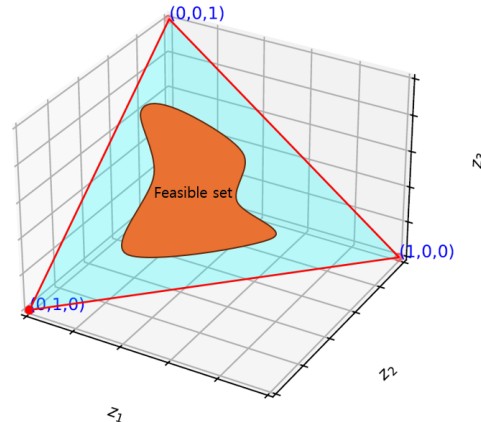

Figure 2: Visualization of the feasible set of policy in the normalized contrastive space. Overall space refers to the contrastive space, and each $z_i$ represents different skills, serving as the axes of this space. The triangle represents the normalized contrastive space. The shaded area within this triangle indicates the feasible set of the policy space. Each point at the vertex of the feasible set represents different skills $z_1, z_2, z_3$.

By ensuring that policies are located at these vertices, we effectively reduce the hypothesis space. This strategic positioning facilitates the identification of distinct policies, thereby enhancing optimization efforts. As described in the Figure 3

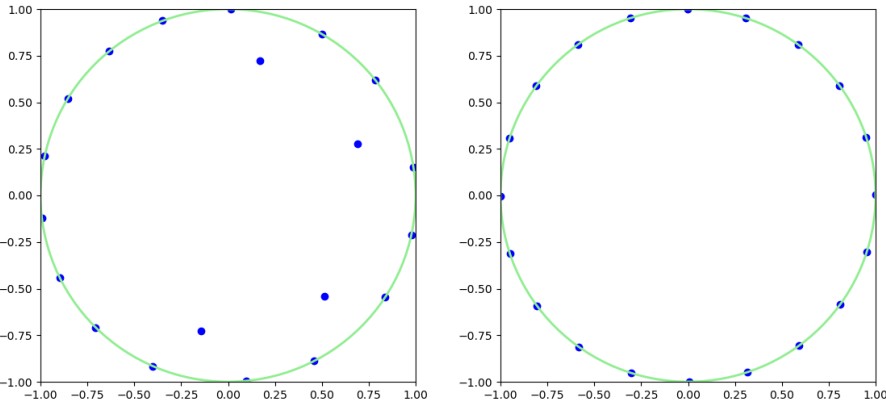

Figure 3: We created 20 skills represented as blue points and maximized the distances between these skills. **[Left]** Maximizing the difference in distribution directly from Formula 6. **[Right]** Result of maximizing the information gain $\mathbb{E}_{z \in Z}[I(\widetilde{\psi}_B(S); Z)_z]$ where $\widetilde{\psi}$ represents normalized version of $\psi$. The average minimum distance $\mathbb{E}_{z \in Z}[\zeta_z]$ of contrastive representation between each pair is 1.9 for the left and 2.1 for the right.

Information gain, in this context, is calculated as the average divergence between each skill's state distribution and the total state distribution, in a manner similar with the approaches used in previous

study (Eysenbach et al., 2022).

$$I(\psi_B(S); Z)_z = D_{KL}\left(\underset{t \in T}{\mathbb{E}}[\mathcal{N}(\psi_B(s_{z,t}), \delta)] \| \underset{z' \in Z}{\mathbb{E}}\left[\underset{t' \in T}{\mathbb{E}}[\mathcal{N}(\psi_B(s_{z',t'}), \delta)]\right]\right)$$

### 4.3 AN INTUITIVE EXAMINATION AND PERFORMANCE GUARANTEE

We investigated how information gain operates in contrastive space and how it manifests in the original space. Initially, similar to the existing algorithm, we modified the embedding function in the information gain from $\widetilde{\psi}_B(S)$ to $\psi_B(S)_z$ to enable faster performance.

**Theorem 4.2.** *(Contrastive space decomposition) The result of the objective function expressed as* $\mathbb{E}_{z \in Z}[I(\widetilde{\psi}_B(S); Z)_z]$ *shares the same goal with the function* $\mathbb{E}_{z \in Z}[I(\psi_B(S)_z; Z)_z]$.

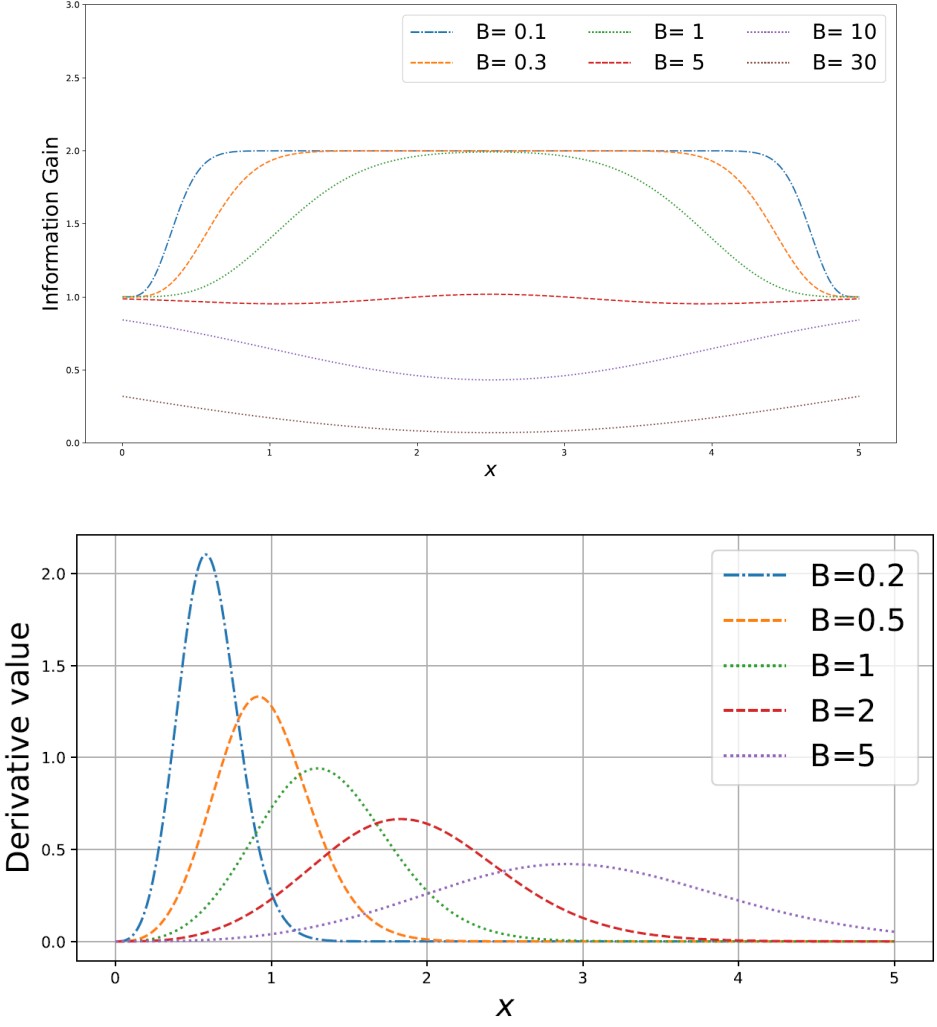

Figure 4: Consider the set $S$ containing pairs of skills. **[up]** The first figure illustrates the information gain from a third skill at $x$ derived from existing two skills at 0 and 5 denoted as $\mathbb{E}_{z \in Z}[I(\psi_B(S = \{0, x, 5\})_z; Z)_z]$ is calculated for different dimension bandwidth B. **[down]** The second figure illustrates the derivative of the information gain between two skills, denoted as $\frac{d}{dx}\mathbb{E}_{z \in Z}[I(\psi_B(S = \{0, x\})_z; Z)_z]$, where the first skill is at position 0 and the second skill occupies position $x$ [down].

As we can see, The repulsive force progressively weakens beyond a certain point, and this point shifts closer to the origin as the bandwidth of the dimension decreases, as shown in Figure 4[down]. Information gain reaches its maximum at the extremes, 0 and 5, but as the dimension bandwidth decreases below a certain level, it starts to peak in the middle in Figure 4[up].

Consider a skill occupying $x_3$ located on the line segment connecting two skills , each occupying $x_1$ and $x_2$. The Information Gain by locating additional skill to $x_3$ exhibits the following properties:

1. As the dimensional bandwidth converges to zero, the skills tend to distribute themselves as equally as possible.

2. When the distance between $x_1$ and $x_2$, denoted as $d(x_1, x_2)$, is less than a threshold $\lambda$, the divergence reaches its minimum at the midpoint of the line segment between $x_1$ and $x_2$, and its maximum at the endpoints $x_1$ and $x_2$.

3. When $d(x_1, x_2) > \lambda$, a concave region begins to emerge at the midpoint between $x_1$ and $x_2$.

**Theorem 4.3.** *(Branching) If the distance between two skills, each occupying a distinct point, exceeds a certain threshold, another skill can be inserted between them such that the information gain of the added skill begins to maximize at the midpoint illustrated in Figure 4. This threshold $\lambda$ can be defined as B as follows:*

$$\text{Given } x = \frac{\lambda}{2}, \quad \frac{d^2}{d^2 x} \mathbb{E}_{z \in Z}[I(\psi_B(S = \{0, x, \lambda\})_z; Z)_z] = 0$$

*when $B = \lambda^2$.*

Now, we consolidates the results of the algorithm's development and derives the performance bounds based on the preceding discussions. Since the policy space cannot be directly measured, our approach differs from existing PAC-MDP studies in that we do not define the bound based on the number of episodes. Instead, given a specific dimensional bandwidth, we provide a performance bound by demonstrating that the space can be covered with a finite number of policies corresponding to that bandwidth.

and based on Theorem 4.1, 4.3, we can derive the general performance bound as follows:

**Theorem 4.4.** *(B-optimality) It is always possible to identify a policy within our optimal policy set such that the performance difference between existing policy and any another optimal policy is less than the maximum performance differences observed among our policies.*

$$|J_\theta(z) - J_{\theta^*}(z)| \leq TR_{max} \cdot \sqrt{\frac{B}{4C}}.$$

In unbounded states, performance differences can increase in proportion to the step size. However, in bounded states, the difference in performance does not necessarily grow with T, and the difference can be expressed using the state size.

## 4.4 OVERALL MECHANISM

In this chapter, we derive an algorithm that actually works based on the proofs mentioned earlier. In the previously mentioned MOMDP, the reward function utilized is of an abstract form employed for theoretical proofs and is not applied in actual training. As we have consistently mentioned, we optimize global policies by maximizing information gain, which serves as the reward in our unsupervised learning approach. The reward function in unsupervised learning adopts the conventional assumption that rewards depend solely on the state. This aligns with the foundational models discussed in prior literature (Eysenbach et al., 2019; Achiam et al., 2018; Lee et al., 2020; Laskin et al., 2022; Eysenbach et al., 2022). Although the time complexity of the KLD operations between Gaussian mixture distributions infinitely increases as time t progresses, we approximate it using k-NN to make it feasible in real-world environments.

**Theorem 4.5.** *(Operationalization) The quantification of information gain for each skill within its respective dimension was accomplished by calculating the KLD between a single Gaussian distribution and a Gaussian mixture distribution. This calculation can be effectively approximated by the following expression:*

$$I(\psi_B(S)_z; Z)_z \propto \frac{1}{T} \sum_{t}^{T} \left( \frac{1}{|N_{knn}(s_{z,t})|} \sum_{s \in N_{knn}(s_{z,t})} \left( 1 - \exp\left( -\frac{(s - s_{z,t})^2}{B} \right) \right)^2 \right)$$

*In this analysis, $N_{knn}$ represents the set of nearest neighbors considered in the computation.*

The following outlines the overall mechanism of our algorithm, which we refer to as the "Concept Block" due to its ability to congregate as needed.

---

**Algorithm 2:** Concept Block (CB)

---

Initialize network $\pi$, dimension Bandwidth B
Set of skils and Initial state $Z, s_0$
Initialize replay buffer D
encodes the skill set Z as K
**while** *not converged* **do**
    t = 0
    $D \leftarrow D \cup (S_0)$
    deploy policys with every skill index at the same time
    **while** $t < T$ **do**
        $A_t \sim \pi_\theta(A_t|S_t; K) + \epsilon$ where $\epsilon \sim \mathcal{N}(0, \sigma^2)$
        $S_{t+1} = T(S_t, A_t)$
        $D \leftarrow D \cup (S_{t+1})$
    **end**
    embed trajactorys $S \in D$ in the contrastive space $\psi_B$
    Update $\pi_\theta$ to maximize information gain $\sum_{s \in N_{knn}} \left( 1 - \exp\left( -\frac{(s - s_z)^2}{B} \right) \right)^2$.
    (by the result of the Operationalization theory.)
**end**

---

## 5 RESULT AND DISCUSSION

We applied this task to the pathfinding domain and measured the progress of each algorithm over a limited number of steps. The evaluation metric in this domain can be interpreted as a separating objective function set, which aims to assign the highest value to a specific policy and the lowest value to others, as distinctly as possible.

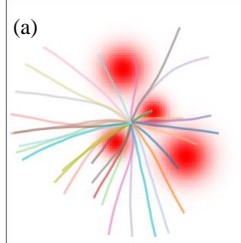 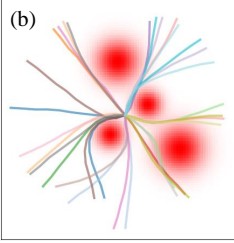 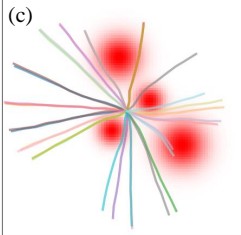 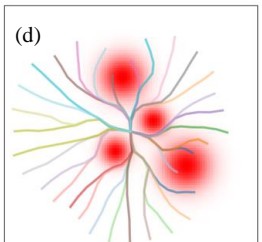

Figure 5: We trained the task of finding maximally distinct paths, starting from the origin and extending in all directions, using contrastive multi-skill reinforcement learning. The closer the agents get to the center (red dot), the slower their movement becomes. Figures (a), (b), (c), and (d) show the learning outcomes of DIAYN, DADS, CIC, and CB, respectively.

Since all four algorithms nearly found the optimal path in short episodes, we extended the episode length, reduced the step size, and set the discount factor below 0.1 to evaluate their performance in environments more similar to realistic scenarios where the step length exceeds 10,000.

As intended by the original paper, DADS prioritized diversity in state-action pairs over the distribution of the states themselves. This resulted in overlapping states but led to the generation of many

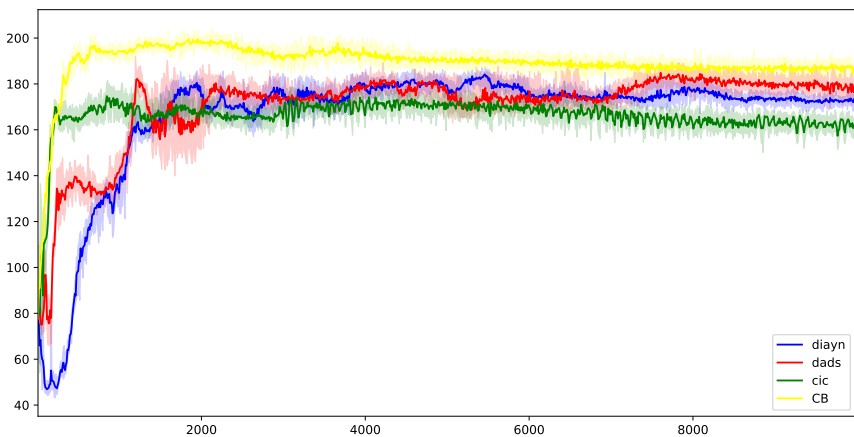

Figure 6: The indices of the skills in Figure 5 are encoded as 5-dimensional vectors consisting of -1 and 1. Since the destination of the paths in the existing algorithms is not uniform, we interpolated these vectors and regenerated the skills based on the extended index. This allowed us to measure the hypervolume (Zitzler & Thiele, 1999), i.e., the volume of the state space covered by the skills. Our model CB demonstrated the best performance from start to finish, while CIC initially showed the next best performance but later declined a bit. DIAYN had the slowest rate of improvement, but it performed better than CIC.

distinct policies. In contrast, CIC, due to the trade-off between maximizing the entropy term and the discriminator's objective, which both aim to diverge early on, failed to discover states behind obstacles. Our algorithm, instead of performing information gain on the states, conducted information gain on the skills. As a result, the distance between identical states varies depending on the perspective of each skill, allowing the algorithm to form a tree structure. This enabled a more uniform exploration of the space and the generation of optimal paths.

Consequently, we investigate that both the hypothesis space optimization is fundamentally driven by curiosity which is formulated as information gain. Preliminary findings indicate that curiosity may play a vital role in achieving optimality. To ensure accurate performance comparisons and to observe the actual process of concept generation, experiments were conducted in relatively simple environments.

By treating knowledge as a distinct dimension, which offers a framework for recognizing other knowledge, we diverge from the traditional approach of utilizing curiosity to reduce the uncertainty of the entire state, which deter the different knowledge to sharing the same pattern in Contrastive Multi-Skill RL domain. This method has successfully facilitated the sharing of common patterns among various knowledge, demonstrating that in our proposed model, maximizing information gain and hypothesis space compression could be identical.

The hallucination issue observed in image generation or LLMs could be actually a phenomenon that arises from the inability to cluster and structure patterns effectively, relying instead on a large number of parameters to train on a case-by-case basis. Hydrogen is the smallest and most fundamental unit among the molecules that make up the world. The word hydrogen originates from the combination of two Greek words: 'hydor' (meaning water) and 'genes' (meaning to generate). In this way, rather than having a hierarchical structure, the knowledge we have acquired helps us acquire new knowledge, and in turn, we use this new knowledge to redefine existing knowledge. Ultimately, knowledge becomes a new dimension that can encompass other knowledge, As learning progresses, it becomes increasingly refined and advanced

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

CONTENTS OF APPENDIX

## A  PRIOR RESEARCH

**Active Learning** Active learning has historically leveraged curiosity in various forms to enhance learning processes. Studies such as (Macedo et al., 2012; Vogl et al., 2019; Schmidhuber, 2007) have utilized discrepancies between predictions and actual outcomes to fuel the learning mechanisms. Additionally, the uncertainty-based methods discussed in (Shankaranarayana, 2023; Howard-Jones & Demetriou, 2009; Zhou & Others, 2020) have been effectively employed to optimize data querying and model training. These methods demonstrate the versatility of curiosity-driven strategies within active learning paradigms, significantly enhancing both the efficiency and accuracy of model development.

**Hypothesis space reduction** Efforts to reduce the hypothesis space in machine learning have led to significant advancements, including methods like meta-learning (Finn et al., 2016; Chen et al., 2016) and transfer learning (Pan & Yang, 2010; Yosinski et al., 2014), which consolidate diverse learning tasks into a unified model framework. Moreover, deep learning integrates extensive parameters to discern complex patterns, necessitating clarity tools such as Explainable AI (Ribeiro et al., 2016; Lundberg & Lee, 2017; Selvaraju et al., 2017). Further innovations like Neural Architecture Search (Zoph & V.Le, 2016), Hyperparameter Optimization (Bergstra & Bengio, 2012), and Automated Machine Learning (Feurer et al., 2016) continue to refine and optimize the hypothesis space from the outset.

**PAC-MDP** The theory of Probably Approximately Correct (PAC) (Valiant, 1984), a pivotal aspect of information theory, has effectively merged the significant trends of curiosity and hypothesis space reduction. This integration has shown that decreasing uncertainty across the entire space and narrowing the hypothesis space increase the likelihood of approximating globally optimal values. In exploring efficient exploration strategies within reinforcement learning, the E3 algorithm (Kearns & Singh, 1998) emerges as a foundational model, particularly for its method of strategically balancing known and unknown state explorations. This approach ensures polynomial time complexity in learning near-optimal policies. Further elaborating on model-based reinforcement learning strategies, the R-max algorithm (Brafman & Tennenholtz, 2002; Kakade, 2003) is distinguished by its methodology, which posits maximal rewards for all unexplored actions. This optimistic initialization facilitates effective exploration, enabling the algorithm to offer near-optimal policies within polynomial

time bounds. MBIE (Strehl & Littman, 2005) improves exploration by using statistical confidence intervals, allowing the algorithm to adapt its exploration based on the uncertainty in the model. This approach provides a more refined balance between exploration and exploitation compared to $R_{\max}$'s optimistic initialization. Delayed Q-learning (Delay-Q) (Strehl et al., 2006) enhances exploration by employing optimistic initialization, like $R_{\max}$, but introduces a delay mechanism that waits until enough evidence is gathered before updating state-action values. The model-free nature of Delay-Q confers significant flexibility, particularly in environments where accurately modeling the dynamics is either challenging or impractical.

## B PROOFS

### B.1 PROOFS IN SECTION 4.1

**Theorem B.1.** *(Density property) The value of $\zeta_{Z^*}$ for an optimal policy set $Z^*$ does not increase upon adding another optimal policy $z_{|Z|+1}$ to the set $Z^*$.*

$$|J_\theta(z) - J_{\theta^*}(z)| \leq |J_\theta(z) - J_\theta(z')|$$

*Proof.* When a new optimal policy, indexed with $|Z| + 1$, is added, the following two conditions are satisfied:

1. Each element of the set $\zeta_{Z^*}$ is either maintained or reduced. Since we already have a set $R^*$ that maximizes the value of $\zeta_{Z^*}$, modifying this set $R*$ would result in a lower value by definition, and even adding new skills would not increase the minimum performance difference $\zeta_z$, which becomes lower. The situation remains the same even if the value of $R*$ is not changed.

2. $\zeta_{|Z|+1}$ has a lower value than each individual element of the existing set $\zeta_Z$. When defining a new optimal policy, indexed with $|Z| + 1$, if the performance difference between this policy and other policies can be made greater than the existing $\zeta_Z$, it implies that one of the existing skills $\pi_i$ could have filled the space where the new policy was added, leading to a higher $\zeta_Z$ with adjusted objective function $R_i$, which would be a contradiction to the definition of the optimal policy set.

Therefore, we can say that these optimal policy set $Z^*$ is $\zeta_{Z^*}$ densely cover the optimal policy space.

$\square$

### B.2 PROOFS IN SECTION 4.3

**Theorem B.2.** *(Contrastive space decomposition) The result of the objective function expressed as $\mathbb{E}_{z \in Z}[I(\widetilde{\psi}_B(S); Z)_z]$ shares the same goal with the function $\mathbb{E}_{z \in Z}[I(\psi_B(S)_z; Z)_z]$.*

*Proof.* This technique, inspired by geometric structures, leverages the fact that the skill parameter $z$ attains its maximum value along the $z$ axis. By maximizing the information gain of skill $z$ along this axis, we can strategically position $z$ at the vertex of the feasible set depicted in Fig. 2. $\square$

In this context, we used the simplified version of the Information Gain derived from 4.2 (**Contrastive space decomposition**). $B$ denotes the dimension bandwidth, $b$ is defined as $\frac{1}{B}$, representing the role of precision. In the calculation of information gain, the variable $\delta$, serving the role of variance in Gaussian distributions, is initially assumed to be sufficiently small to approximate a point distribution. Therefore, in this context, we assume that it decreases in the same manner as $B$. This embedding is illustrated in the graph shown in Figure 7.

**Lemma B.1.** *(Influence Equality)*

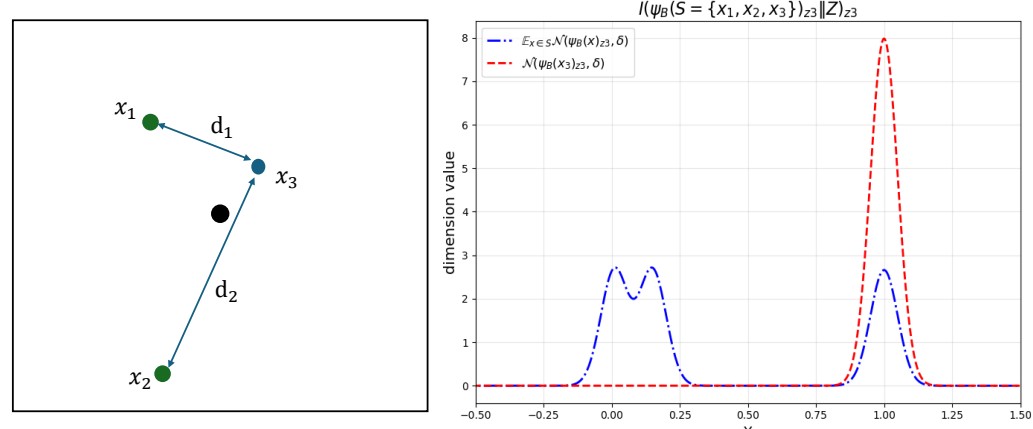

Figure 7: Visualization of how embedding mechanism works. The centers of each component in the Gaussian Mixture Model are $\exp(\frac{-(d_2)^2}{B})$, $\exp(\frac{-(d_1)^2}{B})$, and 1. The skills $z_1$, $z_2$, and $z_3$ occupy the states $x_1$, $x_2$, and $x_3$, respectively.

*Let $\mathcal{N}(\psi(s_{z,t})_z, \delta) = \mathcal{N}(1, \delta)$ and $\mathcal{N}(\psi(s_{z,t})_{z'}, \delta) = \mathcal{N}\left(\frac{1}{T}\sum_{t'}^{T}\exp\left(-\frac{(s_{z,t}-s_{z',t'})^2}{B}\right), \delta\right)$ be two probability distributions. Then*

$$\lim_{B \to 0} \mathrm{D}_{KL}\left(\mathcal{N}(\psi(s_{z,t})_z, \delta) \,\|\, \frac{1}{|Z|}\mathcal{N}(\psi(s_{z,t})_z, \delta) + \frac{1}{|Z|}\sum_{z' \in Z, z' \neq z}\mathcal{N}(\psi(s_{z,t})_{z'}, \delta)\right)$$

$$= \frac{1}{|Z|-1}\sum_{z' \in Z, z' \neq z}\mathrm{D}_{KL}\left(\mathcal{N}(\psi(s_{z,t})_z, \delta) \,\|\, \frac{1}{|Z|}\mathcal{N}(\psi(s_{z,t})_z, \delta) + \frac{|Z|-1}{|Z|}\mathcal{N}(\psi(s_{z,t})_{z'}, \delta)\right)$$

$$= \frac{1}{T(|Z|-1)}\sum_{z' \in Z, z' \neq z}\sum_{t'}^{T}$$

$$\mathrm{D}_{KL}\left(\mathcal{N}(1, \delta) \,\|\, \frac{1}{|Z|}\mathcal{N}(1, \delta) + \frac{|Z|-1}{|Z|}\mathcal{N}\left(\exp\left(-\frac{(s_{z,t}-s_{z',t'})^2}{B}\right), \delta\right)\right)$$

*holds.*

*Proof.* The following is a proof that each center of a previously defined Dirichlet Gaussian Mixture Distribution has an equal influence on the result of the KLD calculation. Define the terms $\alpha$, $\beta$, and $\gamma$ as follows:

$$\alpha = \left(x - \exp\left(-\frac{(s-s')^2}{B}\right)\right)^2, \beta = \left(x - \exp\left(-\frac{(s-s'')^2}{B}\right)\right)^2, \gamma = (1-x)^2$$

Let $M$, $N$, and $L$ be:

$$M = \exp\left(-\frac{\alpha}{B}\right), N = \exp\left(-\frac{\beta}{B}\right), L = \exp\left(-\frac{\gamma}{B}\right)$$

We aim to prove that every element of $s_i$ influences the approximation of the KLD equally, as demonstrated by the following limit:

$$\lim_{B \to 0} \frac{\int L \log\left(\frac{L}{L+M+N}\right) dx - \int L \log\left(\frac{L}{L+2N}\right) dx}{\int L \log\left(\frac{L}{L+M+N}\right) dx - \int L \log\left(\frac{L}{L+2M}\right) dx} = -1$$

This statement holds if the following condition is satisfied:

$$\lim_{B \to 0} \frac{\log(L + M + N) - \log(L + 2M)}{\log(L + M + N) - \log(L + 2N)} = -1$$

To continue the proof, we consider two cases for possible values of $\gamma$.

- **Case 1** ($\gamma \leq \frac{1}{4}$): Assuming $\gamma < \alpha, \beta$,

$$\lim_{B \to 0} \frac{\log(L + M + N) - \log(L + 2M)}{\log(L + M + N) - \log(L + 2N)}$$

$$= \lim_{B \to 0} \frac{\log(1 + \exp\left(\frac{\gamma - \alpha}{B}\right) + \exp\left(\frac{\gamma - \beta}{B}\right)) - \log(1 + 2\exp\left(\frac{\gamma - \alpha}{B}\right))}{\log(1 + \exp\left(\frac{\gamma - \alpha}{B}\right) + \exp\left(\frac{\gamma - \beta}{B}\right)) - \log(1 + 2\exp\left(\frac{\gamma - \beta}{B}\right))} \quad \text{(Simplification)}$$

$$= \lim_{B \to 0} \frac{\exp\left(\frac{\gamma - \alpha}{B}\right) + \exp\left(\frac{\gamma - \beta}{B}\right) - 2\exp\left(\frac{\gamma - \alpha}{B}\right)}{\exp\left(\frac{\gamma - \alpha}{B}\right) + \exp\left(\frac{\gamma - \beta}{B}\right) - 2\exp\left(\frac{\gamma - \beta}{B}\right)} \quad \text{(by Taylor expansion)}$$

$$= -1$$

- **Case 2** ($\gamma > \frac{1}{4}$): Assuming $\exists c > 0$, such that $\gamma - c > \alpha, \beta$, we have

$$\lim_{B \to 0} \frac{\log(L + M + N) - \log(L + 2M)}{\log(L + M + N) - \log(L + 2N)}$$

$$= \lim_{B \to 0} \frac{\log(M + N) - \log(2M)}{\log(M + N) - \log(2N)} \quad \text{(Simplification)}$$

$$= \lim_{b \to \infty} \frac{\log(\exp(-\alpha b) + \exp(-\beta b)) - \log(2\exp(-\alpha b))}{\log(\exp(-\alpha b) + \exp(-\beta b)) - \log(2\exp(-\beta b))} \quad \text{(l'Hôpital's rule)}$$

$$= \lim_{b \to \infty} \frac{(\alpha + \alpha' b)\exp(-\beta b) - (\beta + \beta' b)\exp(-\beta b)}{-(\alpha + \alpha' b)\exp(-\alpha b) + (\beta + \beta' b)\exp(-\alpha b)}$$

$$= \lim_{b \to \infty} -\exp((\alpha - \beta)b)$$

$$= \lim_{B \to 0} -\exp\left(\frac{\alpha - \beta}{B}\right) \quad \text{(Change of variable)}$$

$$= -1$$

The last equality holds as

$$\lim_{B \to 0} \left(\exp\left(-\frac{(s - s'')^2}{B}\right) - \exp\left(-\frac{(s - s')^2}{B}\right)\right) / B = 0.$$

$\square$

**Lemma B.2.** *(Natural component disengagement) During the computation of the KLD between a single Gaussian distribution and a mixture of Gaussian distributions, if the variance of gaussian distribution $\delta$ is small enough, the influence of the dominant Gaussian component in the mixture becomes overwhelmingly large compared to other, leading to a cancellation of the first term on the right side of the divergence equation:*

$$\lim_{B \to 0} D_{KL}(\mathcal{N}(\psi_B(s_{z,t})_z, \delta) \| \frac{1}{K}\mathcal{N}(\psi_B(s_{z,t})_z, \delta) + \frac{K - 1}{K}\mathcal{N}(\psi_B(s_{z,t})_{z'}, \delta))$$

$$= D_{KL}(\mathcal{N}(\psi_B(s_{z,t})_z, \delta) \| \mathcal{N}(\psi_B(s_{z,t})_{z'}, \delta))$$

*Proof.* Let $M = \exp\left(-\frac{(1 - x)^2}{\delta}\right)$ and $N = \exp\left(-\frac{(\exp(-\alpha b) - x)^2}{\delta}\right)$ be two functions of $x$. Now, consider the limit:

$$\lim_{B \to 0} D_{KL}(\mathcal{N}(\psi_B(s_{z,t})_z, \delta) \| \frac{1}{K}\mathcal{N}(\psi_B(s_{z,t})_z, \delta) + \frac{K - 1}{K}\mathcal{N}(\psi_B(s_{z,t})_{z'}, \delta))$$

$$= D_{KL}(\mathcal{N}(\psi_B(s_{z,t})_z, \delta) \| \mathcal{N}(\psi_B(s_{z,t})_{z'}, \delta))$$

This is proportional to:

$$\lim_{b\to\infty} \int M \log \frac{M}{\frac{1}{Z}M + \frac{Z-1}{Z}N} = \lim_{b\to\infty} \int_{1-\epsilon}^{1+\epsilon} M \log \frac{M}{\frac{1}{Z}M + \frac{Z-1}{Z}N}$$

$$= \lim_{b\to\infty} \int_{1-\epsilon}^{1+\epsilon} M \log \frac{M}{N}$$

where the last equaliti holds, as for all $x \in (1 - \epsilon, 1 + \epsilon)$, we have:

$$\lim_{b\to\infty} -\left(\exp(-\alpha b) - x\right)^2 \delta = -\left(x\right)^2 \delta < -(1-x)^2\delta.$$

Since the simplified quantity is proportional to

$$\mathrm{D}_{KL}(\psi_B(s_{z,t})_z \,\|\, \psi_B(s_{z,t})_{z'}).$$

we conclude the proof.

$\square$

**Lemma B.3.** *(Natural gradient vanishing) As the squared distance between two states $s$ and $s'$ approaches infinity, the derivative of the KL divergence between their associated Gaussian distributions approaches zero, thereby rendering its influence negligible. Mathematically,*

$$\lim_{(x-x')^2\to\infty} \frac{d}{dx'}\mathrm{D}_{KL}(\mathcal{N}(1, B) \,\|\, \mathcal{N}(\exp(-\frac{(x-x')^2}{B}), B)) = 0$$

*Proof.* The derivative of the KL divergence with respect to $x'$ is given by:

$$\frac{d}{dx'}\mathrm{D}_{KL}(\mathcal{N}(1, B) \,\|\, \mathcal{N}(\exp(-\frac{(x-x')^2}{B}), B))$$

$$= \frac{d}{dx'} \left(1 - \exp\left(-\frac{(x-x')^2}{B}\right)\right)^2 /(2B)$$

$$= \left((-(2x' - 2x)/B) \exp\left(-\frac{(x-x')^2}{B}\right) + (-2(2x' - 2x)/B) \exp\left(-\frac{2(x-x')^2}{B}\right)\right) /(2B)$$

$$= \exp\left(-(x-x')^2/B\right) \left(1 + 2\exp\left(-(x-x')^2/B\right)\right) (x-x')/B^2$$

By taking the limit,

$$\lim_{(x-x')^2\to\infty} \exp\left(-(x-x')^2/B\right) (x-x')/B^2 = 0$$

we conclude the proof. $\square$

**Theorem B.3.** *(Branching) If the distance between two skills, each occupying a distinct point, exceeds a certain threshold, another skill can be inserted between them such that the information gain of the added skill begins to maximize at the midpoint illustrated in Figure 4. This threshold $\lambda$ can be defined as B as follows:*

$$Given \ x = \frac{\lambda}{2}, \quad \frac{d^2}{d^2x} \mathop{\mathbb{E}}_{z\in Z}[I(\psi_B(S = \{0, x, \lambda\})_z; Z)_z] = 0$$

*when $B = \lambda^2$.*

*Proof.* The second derivative of the equation for information gain with respect to x can be expressed separately as follows:

$$\frac{d^2}{d^2x} \left(D_{kl}(\mathcal{N}(\psi_B(x), \delta) \,|\, \mathcal{N}(\psi_B(\lambda)), \delta) + D_{kl}(\mathcal{N}(\psi_B(x), \delta) \,|\, \mathcal{N}(\psi_B(0)), \delta)\right) = 0$$

The concave region of information gain begins at the midpoint between two states. Therefore, we will determine whether the function is concave or convex when $x$ is $\frac{\lambda}{2}$ and identify the transition condition when it switches from convex to concave.

$$
\frac{d}{dx}\left(D_{kl}(\mathcal{N}(\psi_B(x),\delta) \mid \mathcal{N}(\psi_B(\lambda)),\delta) + D_{kl}(\mathcal{N}(\psi_B(x),\delta) \mid \mathcal{N}(\psi_B(0)),\delta)\right)
$$

$$
= \frac{d}{dx}\left(\frac{(1-\exp(-x^2/B))^2}{\delta} + \frac{(1-\exp(-(\lambda-x)^2/B))^2}{\delta}\right)
$$

$$
= \frac{d}{dx}\frac{1}{\delta}\left(\exp(-\frac{2x^2}{B}) - 2\exp(-\frac{x^2}{B}) + \exp(-\frac{2(\lambda-x)^2}{B}) - 2\exp(-\frac{(\lambda-x)^2}{B})\right)
$$

$$
= \frac{-4x}{B\delta}\exp(-\frac{2x^2}{B}) + \frac{4x}{B\delta}\exp(-\frac{x^2}{B})
$$

$$
+ \frac{-4(x-\lambda)}{B\delta}\exp(-\frac{2(\lambda-x)^2}{B}) + \frac{4(x-\lambda)}{B\delta}\exp(-\frac{(\lambda-x)^2}{B})
$$

Differentiating the simplified expression again:

$$
= -x\exp(-\frac{2x^2}{B}) + x\exp(-\frac{x^2}{B}) - (x-\lambda)\exp(-\frac{2(\lambda-x)^2}{B}) + (x-\lambda)\exp(-\frac{(\lambda-x)^2}{B}).
$$

$$
\frac{d}{dx}\left(-x\exp(-\frac{2x^2}{B}) + x\exp(-\frac{x^2}{B}) - (x-\lambda)\exp(-\frac{2(\lambda-x)^2}{B}) + (x-\lambda)\exp(-\frac{(\lambda-x)^2}{B})\right)
$$

$$
= -\exp(-\frac{2x^2}{B}) + \exp(-\frac{x^2}{B}) - \exp(-\frac{2(x-\lambda)^2}{B}) + \exp(-\frac{(x-\lambda)^2}{B})
$$

$$
+ x\left(\frac{4x}{B}\exp(-\frac{2x^2}{B}) - \frac{2x}{B}\exp(-\frac{x^2}{B})\right)
$$

$$
= +(x-\lambda)\left(\frac{4(x-\lambda)}{B}\exp(-\frac{2(x-\lambda)^2}{B}) - \frac{2(x-\lambda)}{B}\exp(-\frac{(x-\lambda)^2}{B})\right)
$$

$$
- 2\exp(-\frac{\lambda^2}{2B}) + 2\exp(-\frac{\lambda^2}{4B}) + \lambda\left(\frac{2\lambda}{B}\exp(-\frac{\lambda^2}{2B}) - \frac{\lambda}{B}\exp(-\frac{\lambda^2}{4B})\right).
$$

After deleting constant terms:

$$
- 2\exp(-\frac{\lambda^2}{4B}) + 2 + \lambda\left(\frac{2\lambda}{B}\exp(-\frac{\lambda^2}{4B}) - \frac{2\lambda}{B}\right) = 0 \text{ when } B = \lambda^2.
$$

The influence of states further than this nearest state decreases exponentially as the dimension bandwidth reduces, according to Lemma B.3. $\qquad\square$

**Lemma B.4.** *(Inequality in Kullback-Leibler divergence(KLD) for low-variance Gaussian Mixture Models) Assume two arbitrary Gaussian mixture models $\mathcal{M}_1$ and $\mathcal{M}_2$ comprising low-variance components, and $M_1$, $M_2$ the number of modes in the Gaussian mixture distribution. As $\sigma \to 0$, the models can be expressed as:*

$$
\mathcal{M}_1 = \frac{1}{M_1}\sum_{i=1}^{M_1}\mathcal{N}(\mu_i,\sigma^2)
$$

$$
\mathcal{M}_2 = \frac{1}{M_2}\sum_{j=1}^{M_2}\mathcal{N}(\nu_j,\sigma^2)
$$

*where $\mu_i$ and $\nu_j$ are the means of the Gaussian components in $M_1$ and $M_2$ respectively.*

*Furthermore, the KLD between these models is bounded as follows:*

$$D_{KL}(\mathcal{M}_1\|\mathcal{M}_2) \leq \frac{1}{M_1 M_2}\sum_{i=1}^{M_1}\sum_{j=1}^{M_2} D_{KL}(\mathcal{N}(\mu_i,\sigma^2)\|\mathcal{N}(\nu_j,\sigma^2)).$$

*Proof.* Consider three Gaussian distributions:

$$P_X := \mathcal{N}(\alpha,\sigma^2), \quad P_Y := \mathcal{N}(\beta,\sigma^2), \quad P_Z := \mathcal{N}(\gamma,\sigma^2)$$

As $\sigma$ approaches zero, the KLD is predominantly defined by the values near the local maxima of the first term. The divergence can be expressed by the following integrals:

$$\lim_{\sigma\to 0}\int_{-\infty}^{\infty}\left(\exp\left(-\frac{(x-\alpha)^2}{\sigma^2}\right)+\exp\left(-\frac{(x-\beta)^2}{\sigma^2}\right)\right)$$
$$\cdot\log\left(\frac{\exp\left(-\frac{(x-\alpha)^2}{\sigma^2}\right)+\exp\left(-\frac{(x-\beta)^2}{\sigma^2}\right)}{2\exp\left(-\frac{(x-\gamma)^2}{\sigma^2}\right)}\right)dx$$

$$=\lim_{\sigma\to 0}\left[\int_{\alpha-\epsilon}^{\alpha+\epsilon}\exp\left(-\frac{(x-\alpha)^2}{\sigma^2}\right)\log\left(\frac{\exp\left(-\frac{(x-\alpha)^2}{\sigma^2}\right)+\exp\left(-\frac{(x-\beta)^2}{\sigma^2}\right)}{2\exp\left(-\frac{(x-\gamma)^2}{\sigma^2}\right)}\right)dx\right.$$

$$\left.+\int_{\beta-\epsilon}^{\beta+\epsilon}\exp\left(-\frac{(x-\beta)^2}{\sigma^2}\right)\log\left(\frac{\exp\left(-\frac{(x-\alpha)^2}{\sigma^2}\right)+\exp\left(-\frac{(x-\beta)^2}{\sigma^2}\right)}{2\exp\left(-\frac{(x-\gamma)^2}{\sigma^2}\right)}\right)dx\right]$$

$$=\lim_{\sigma\to 0}\left[\int_{\alpha-\epsilon}^{\alpha+\epsilon}\exp\left(-\frac{(x-\alpha)^2}{\sigma^2}\right)\log\left(\frac{\exp\left(-\frac{(x-\alpha)^2}{\sigma^2}\right)}{2\exp\left(-\frac{(x-\gamma)^2}{\sigma^2}\right)}\right)dx\right.$$

$$\left.+\int_{\beta-\epsilon}^{\beta+\epsilon}\exp\left(-\frac{(x-\beta)^2}{\sigma^2}\right)\log\left(\frac{\exp\left(-\frac{(x-\beta)^2}{\sigma^2}\right)}{2\exp\left(-\frac{(x-\gamma)^2}{\sigma^2}\right)}\right)dx\right]$$

$\therefore$ the limit as $\sigma\to 0$, $2D_{KL}\left(\frac{P_X+P_Y}{2}\|P_Z\right)$ equals the sum of the divergences of $P_Y$ and $P_X$ from $P_Z$:

$$\lim_{\sigma\to 0}\left[D_{KL}(P_Y\|P_Z)+D_{KL}(P_X\|P_Z)\right]$$

Given positive constants $A, B$, and $C$ such that $A, B, C > 0$, we consider the following logarithmic inequalities:

$$2\log\frac{A+B}{2} \geq \log A + \log B,$$
$$2\log C - 2\log\frac{A+B}{2} \leq 2\log C - (\log A + \log B),$$
$$2C\log\frac{C}{\left(\frac{A+B}{2}\right)} \leq C\log\frac{C}{A} + C\log\frac{C}{B}.$$

These inequalities are derived from the concavity of the logarithmic function and are used to demonstrate the following inequality involving KLD for any probability distribution $P$:

$$\therefore 2D_{KL}\left(P\|\frac{P_X+P_Y}{2}\right) \leq D_{KL}(P\|P_X) + D_{KL}(P\|P_Y)$$

Using the following foundational results:

- The limit of twice the KLD for a bimodal Gaussian distribution as $\sigma \to 0$ equals the sum of the divergences of each component from a reference model $P_Z$:

$$\therefore \lim_{\sigma \to 0} 2D_{KL}\left(\frac{P_X + P_Y}{2}\|P_Z\right) = \lim_{\sigma \to 0}[D_{KL}(P_Y\|P_Z) + D_{KL}(P_X\|P_Z)]$$

- The KLD for any distribution $P$ with respect to the midpoint of distributions $P_X$ and $P_Y$ is bounded by the sum of the divergences to each distribution:

$$\therefore 2D_{KL}\left(P\|\frac{P_X + P_Y}{2}\right) \le D_{KL}(P\|P_X) + D_{KL}(P\|P_Y)$$

We can derive an inequality for Gaussian mixture models as follows:

$$M_{1,i} = \mathcal{N}(\mu_i, \sigma^2), \quad M_{2,j} = \mathcal{N}(\nu_j, \sigma^2)$$

$$\mathcal{M}_1 = \frac{1}{|M_1|}\sum_{i=1}^{|M_1|}\mathcal{N}(\mu_i, \sigma^2), \quad \mathcal{M}_2 = \frac{1}{|M_2|}\sum_{j=1}^{|M_2|}\mathcal{N}(\nu_j, \sigma^2)$$

where $\sigma \to 0$, and where $\mu_i$ and $\nu_j$ are the means of the Gaussian components in $\mathcal{M}_1$ and $\mathcal{M}_1$, respectively.

The KLD between these models is bounded by the average of the pairwise divergences between individual components:

$$D_{KL}(\mathcal{M}_1\|\mathcal{M}_2) \le \frac{1}{|M_1| \cdot |M_2|}\sum_{i=1}^{|M_1|}\sum_{j=1}^{|M_2|} D_{KL}(M_{1,i}\|M_{2,j})$$

$\square$

First, we can derive the performance bound from the definition as follows:

**Lemma B.5.** *(Performance bound among optimal skills)Let $z, z' \in Z$ with $z \ne z'$, policy space $\Theta$. The $J_\theta(z)$ is upper bounded by*

$$|J_\theta(z) - J_\theta(z')| \le R_{max}\sqrt{\frac{1}{2}\sum_{t=1}^{T}\sum_{t'=1}^{T}\frac{d(s_{z,t}, s_{z',t'})^2}{2C}}.$$

*Proof.* The following derivation shows how the upper bound for the average difference of performance between two policies indexed as $z$ and $z'$ can be computed. This upper limit is influenced by the total variation and KLD between the policies' distributions over states.

$$|J_\theta(z) - J_\theta(z')| \le \mathbb{E}_{s \sim S}[\|\mu_{z,\theta}(s)O'_z(s) - \mu_{z',\theta}(s)O'_z(s)\|] \tag{7}$$

$$\le (TR_{max})\int_S \|\mu_{z,\theta}(s) - \mu_{z',\theta}(s)\|ds \tag{8}$$

$$\le (TR_{max})\sqrt{\frac{1}{2}D_{KL}(\mu_{z,\theta}(s)\|\mu_{z',\theta}(s))} \quad \text{(by Pinsker's Inequality)} \tag{9}$$

$$\le (TR_{max})\sqrt{\frac{1}{2T^2}\sum_{t=1}^{T}\sum_{t'=1}^{T}D_{KL}(\mathcal{N}(x|s_{z,t}, C \cdot E)\|\mathcal{N}(x|s_{z',t'}, C \cdot E))} \tag{10}$$

$$\le (TR_{max})\sqrt{\frac{1}{2T^2}\sum_{t=1}^{T}\sum_{t'=1}^{T}\frac{d(s_{z,t}, s_{z',t'})^2}{2C}} \tag{11}$$

Inequality (7) directly cames out from the definition of J, inequality (8) cames from the definition of $R_{max}$, (10) derived (by Lemma B.4), (11) derived from the definition of kld between single gaussian distributions $\square$

**Theorem B.4.** *(B-optimality) It is always possible to identify a policy within our optimal policy set such that the performance difference between existing policy and any another optimal policy is less than the maximum performance differences observed among our policies:*

$$|J_\theta(z) - J_{\theta^*}(z)| \le TR_{max} \cdot \sqrt{\frac{B}{4C}}.$$

*Proof.*

$$|J_\theta(z) - \max_{\theta' \in \Theta} J_{\theta'}(z)| \le |J_\theta(z) - J_\theta(z')| \tag{12}$$

$$\le (TR_{\max})\sqrt{\frac{1}{2T^2} \sum_t^T \sum_{t'}^T \frac{d(s_{z,t}, s_{z',t'})^2}{2C}} \tag{13}$$

$$= (TR_{\max})\sqrt{\frac{B}{4C}}. \tag{14}$$

By applying Theorem 4.1 for (12), applying Lemma B.5 for (13), applying Theorem 4.3 for (14), we conclude the proof. □

### B.3 PROOFS IN SECTION 4.4

**Theorem B.5.** *(Operationalization) The quantification of information gain for each skill within its respective dimension was accomplished by calculating the KLD between a single Gaussian distribution and a Gaussian mixture distribution. This calculation can be effectively approximated by the following expression:*

$$I(\psi_B(S)_z; Z)_z \propto \frac{1}{T} \sum_t^T \left( \frac{1}{|N_{knn}(s_{z,t})|} \sum_{s \in N_{knn}(s_{z,t})} \left( 1 - \exp\left( -\frac{(s - s_{z,t})^2}{B} \right) \right)^2 \right)$$

*In this analysis, $N_{knn}$ represents the set of nearest neighbors considered in the computation.*

*Proof.* from the definition,

$$I(\psi_B(S)_z; Z)_z = D_{KL}(\underset{t \in T}{\mathbb{E}}[\mathcal{N}(\psi_B(s_{z,t})_z, \delta)] \| \underset{z' \in Z}{\mathbb{E}}[\underset{t' \in T}{\mathbb{E}}[\mathcal{N}(\psi_B(s_{z',t'})_z, \delta)]])$$

Using the facts

$$\mathcal{N}(\psi(s_{z,t})_z, \delta) = \mathcal{N}(1, \delta), \quad \mathcal{N}(\psi(s_{z,t})_{z'}, \delta) = \mathcal{N}\left( \frac{1}{T} \sum_{t' \in T} \exp\left( -\frac{(s_{z',t'} - s_{z,t})^2}{B} \right), \delta \right)$$

The objective Function of proposed method is given by

$$\mathrm{D}_{KL}\left( \mathcal{N}(1, \delta) \, \| \, \frac{1}{|Z|}\mathcal{N}(1, \delta) + \frac{1}{|Z|} \sum_{z' \in Z, z' \ne z} \frac{1}{T} \sum_{t \in T} \mathcal{N}\left( \frac{1}{T'} \sum_{t' \in T'} \exp\left( -\frac{(s_{z',t'} - s_{z,t})^2}{B} \right), \delta \right) \right)$$

Applying Lemmas 4.1, 4.2, we approximate the KL divergence in the limit as $B$ approaches zero:

$$= \underset{t \in T}{\mathbb{E}}\left( \frac{1}{|Z|} \sum_{z' \in Z, z' \ne z} \underset{t' \in T'}{\mathbb{E}} \left( D_{\mathrm{KL}}\left( \mathcal{N}(1, \delta) \, \| \, \mathcal{N}\left( \exp\left( -\frac{(s_{z',t'} - s_{z,t})^2}{B} \right), \delta \right) \right) \right) \right)$$

The KL divergence between two Gaussian distributions is expressed as follows:

$$D_{\mathrm{KL}}(\mathcal{N}(a, B) \| \mathcal{N}(b, B)) = \frac{(a - b)^2}{2B}.$$

Thus, Applying Lemma 4.3, the formula representing the information gain for the skill z is:

$$\frac{1}{T}\sum_t^T \frac{1}{|N_{knn}(s_{z,t})|} \sum_{s \in N_{knn}(s_{z,t})} \frac{(1 - \exp(-\frac{(s-s_{z,t})^2}{B}))^2}{2B}.$$

Since all skills are divided by the same constant 2B, we conclude the proof by eliminating this term

$\square$

## C  DETAILED ANALYSIS IN EXPERIMENT

### C.1  EXPERIMENTAL SETTING

We employed the policy gradient method to operate within a continuous state and action environment. Each skill was initiated from a central point and allowed to move in random directions. A two-layered single Multi-Layer Perceptron (MLP) was used. As we trained 32 skills, we encoded the index of each skill using a 5-digit array composed of -1 or 1, reflecting that $2^5 = 32$ For interpolation, we paired the two closest skills and divided their indices by 20. For example, if the indices of the two closest policies were $(-1, 1, 1, -1, 1)$ and $(-1, 1, 1, 1, -1)$, the result of the interpolation would be $(-1, 1, 1, -0.9, 0.9) \cdots (-1, 1, 1, 0.9, -0.9)$. The input tensor is a concatenation of the batched indices and the batched state. The scalar value of the action ranges from $[0, 1]$, with no constraints on the direction.

### C.2  ENVIRONMENT DESCRIPTION

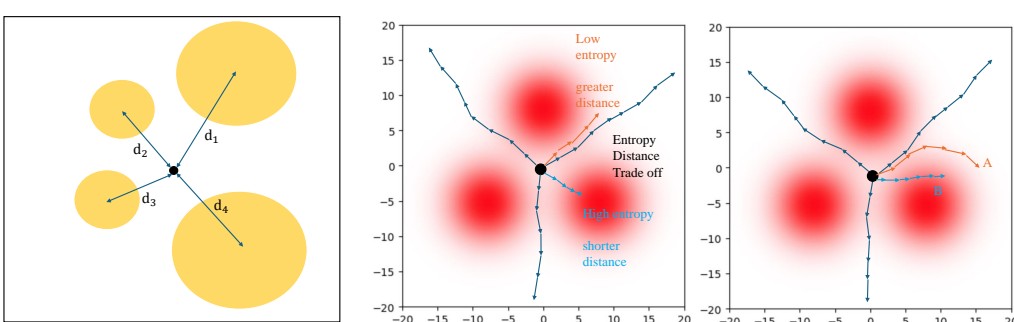

Figure 8: **[Left]** Illustration of the points generating mechanism, which serves as the center of each friction zone. The distance $d$ is selected from the negative binomial distribution $d \sim \mathrm{NB}(r, p)$ where $r = 2$, $p = 0.1$. Constraint: $d_i > 2$ and the mean of each center $m_i$ satisfying $|m_i - m_j| > \frac{d_i + d_j}{2}$. **[Mid]** We intentionally created trade-off scenarios in the design of our algorithm. **[Right]** Our algorithm tries to find the optimal path while congregating the policies as much as they need, whereas the existing algorithm tends to select shorter distances in its attempt to maximize entropy, which consequently results in a decrease in optimality.

When generating the map environment, our goal was not to impose barriers that restrict movement but rather to allow the agent to autonomously select actions to find the optimal path. Analogous to how movement is typically slower in mountainous areas and faster on flat terrain, we crafted a 2D map featuring regions that decelerate movement. These slower regions were defined around sampled points, with friction values assigned using the function $\exp\left(-\frac{d^2}{\xi}\right)$, where $d$ denotes the distance from the center of each region and $\xi$ represents the variance.

the Coefficient Of Friction(COF) can be expressed as below:

$$\left( \sum_{i=1}^{n} \exp\left( -\frac{(m_x - m_i)^2}{\xi_i} \right) + \sum_{i=1}^{n} \exp\left( -\frac{(m_{x'} - m_i)^2}{\xi_i} \right) \right)$$

The resulting action is then adjusted based on the COF as follows:

$$\text{final action} = (1 - \frac{\text{COF}}{2}) \times \text{action}$$

As demonstrated in Figure 8, our approach aimed to establish a trade-off between entropy and distance. Contrary to previous information gain-based skill discovery studies that inherently linked increases in entropy to corresponding increases in distance, Our research indicates that the algorithm is capable of finding novel states, even in the context of decreasing entropy

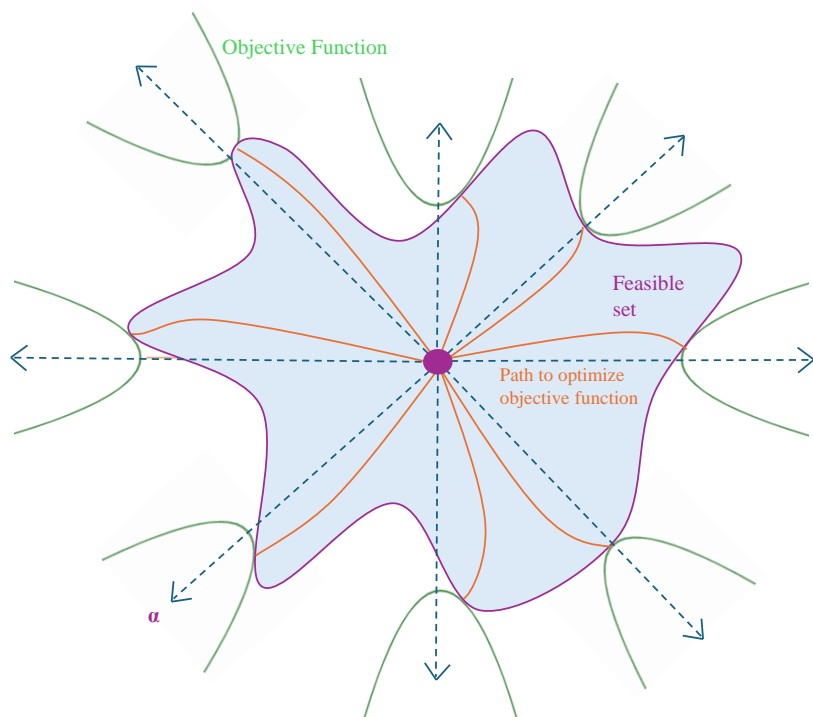

Figure 9: We have defined a set of objective functions consistant to prior definition, $\mathcal{O} = \{\mathbf{O} \mid \mathbf{O} : \mathcal{S} \to \mathbb{R}\}$ and each of objective functions are defined as $O_z(x) = |x \cdot \alpha_z| - |x \times \alpha_z|^2$, where x denotes the location of the state and $\alpha_z$ denotes the angle of zth objective function. we selected the $z$ most distinct ones to satisfies the **separating objective function set**. We then chose $z$ policies that maximize the given z objective function from the policy set which were initially trained to maximize $\zeta_z$. During training, the policies operated probabilistically, but for simplification in rendering, the policies were made to operate deterministically, resulting the value of $\gamma$ to 1 in formula (5). This resulted in only the final state of the trajectory influencing the performance, with the effectiveness of each policy determined by how well the position of the last state maximized the objective function. the terminate state conducted from our model is illustrated as Feasible set in this figure, mathmetically fomulated as Constraint $|x_\alpha| \leq M(\alpha)$, where $|x_\alpha|$ denotes the distance from the origin with respect to angle $\alpha$ and $M(\alpha)$ means the maximal distance generated my our model with respect to the angle $\alpha$.

## C.3 EVALUATION METRIC

We estimated the **HyperVolume** (Zitzler & Thiele, 1999) in the Figure 6 using the following formula. We calculated $d_i$ as the maximum distance achieved by our policy, which was selected based on the mechanism described in Figure 9.

$$\sum_{i=1}^{n} d_{mod(i,n)} \cdot d_{mod(i+1,n)} \cdot \sin(|\omega_{mod(i,n)} - \omega_{mod(i+1,n)}|).$$

## C.4 ADDITIONAL RESULTS

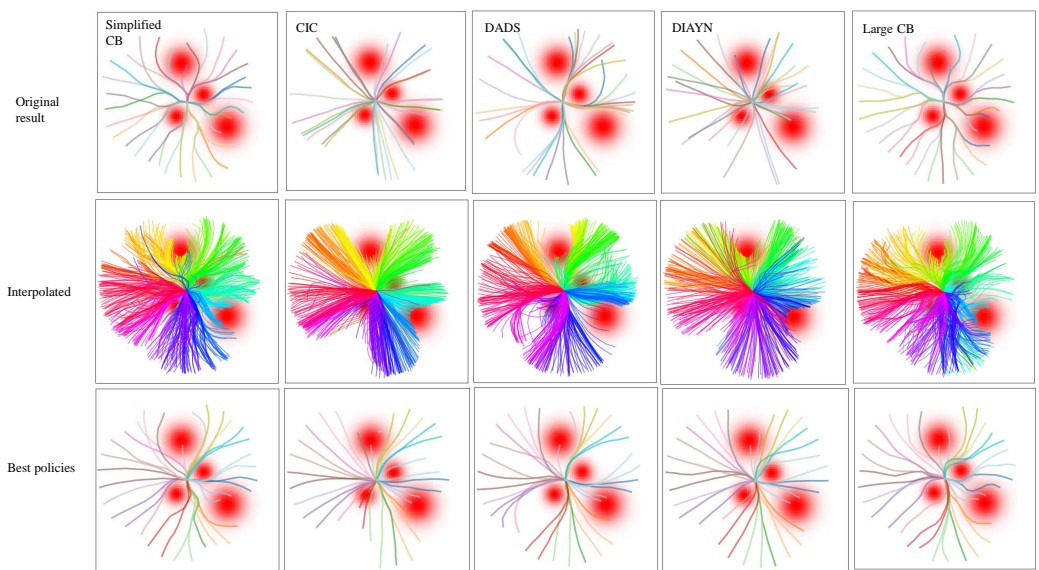

Figure 10: The figures describe the process of measuring performance. The top row shows the results after initially training with 32 skills. The middle row depicts the interpolation process, which involves rendering by dividing the index between the two nearest skills into 20 segments. The bottom row illustrates the selection of policies that maximize a given set of 33 different Separating Objective Function Sets, based on these results.

Figure 10 provides a detailed description of our performance measurement approach. In practice, when utilizing multi-skill reinforcement learning, we extend beyond the trained policy. We employ a gradient-based method to identify and implement a policy that optimally adjusts to the specified object. Moreover, in techniques like CIC and DADS, the focus is on exploring the most distant areas, which results in an average path length that exceeds our method. However, the substantial number of still unexplored areas necessitates this approach to accurately assess performance. The objective function used for training the large CB is given by

$$\mathbb{E}_{z \in Z}[I(\widetilde{\psi}_B(S); Z)_z],$$

and the objective function used for training the simplified CB, which is result of Theorem (**Contrastive space decomposition**) is

$$\mathbb{E}_{z \in Z}[I(\psi_B(S)_z; Z)_z].$$

Figure 11: This is a TensorBoard screenshot showing the time per epoch required to train the above five models. The model that took the longest time is the large CB, followed by CIC, DADS, Simplified CB, and DIAYN in that order.

As illustrated in Figure 11, while the actual outcomes of the Large CB and Simplified CB are identical, a variance in training speed is evident. The reason for this discrepancy lies in the structure of the large CB; as the number of skills $|Z|$ increases, so too does the quantity of features per skill, necessitating the calculation of distances between features across all skill pairs. This results in a time complexity that escalates proportionally to $|Z|^3$. Conversely, the simplified CB merely requires the computation of rewards pertinent to the dimensions engaged by the updated skill, leading to a time complexity that grows with $|Z|^2$, which is analogous to that of the entropy bonus term in CIC.

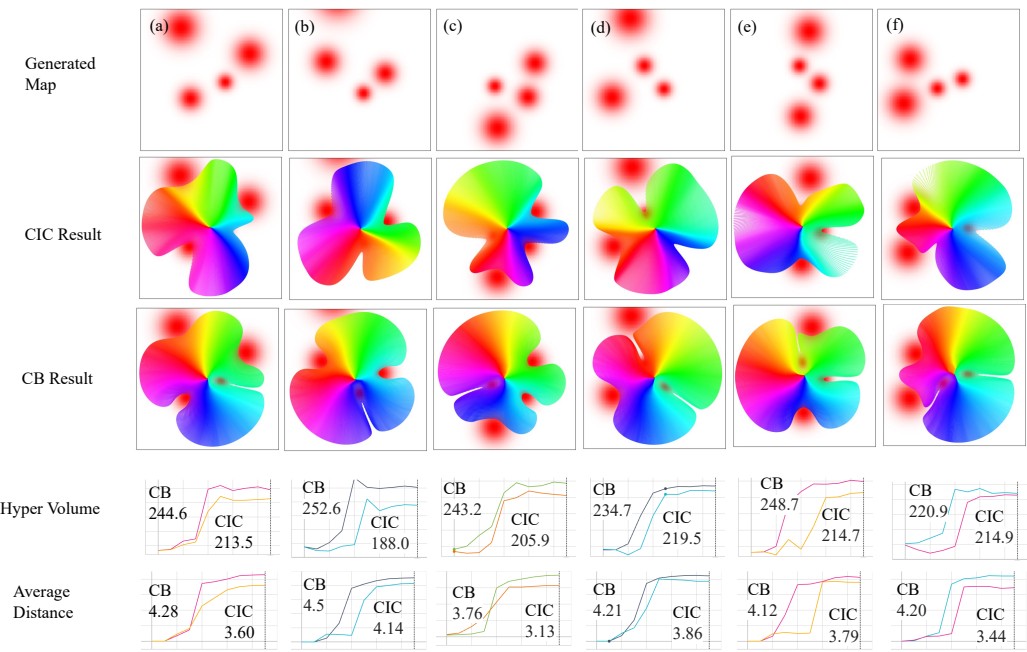

Figure 12: Six maps were generated using the specified map generation mechanism and trained with both CIC and CB algorithms. For each training iteration, 32 skills were randomly sampled from a 2D continuous skill space. The k-nearest neighbors (k-NN) parameter was set to 100 for both algorithms. The two bottom rows contain a total of 12 line graphs, with the horizontal axes displayed on a logarithmic scale to illustrate iterations.

| Improvement | case(a) | case(b) | case(c) | case(d) | case(e) | case(f) | Average |
|---|---|---|---|---|---|---|---|
| Volume | 14.5% | 34.3% | 18.1% | 6.9% | 15.8% | 2.7% | 15.4% |
| Distance | 18.8% | 8.6% | 20.1% | 9.0% | 8.7% | 22.0% | 14.5% |

Table 1: Illustration of the enhanced performance of CB compared to CIC.

The original paper on CIC does not fix the indices of the policies to be trained; instead, it selects them randomly at each iteration. However, when using this method in CIC, if the indices are high-dimensional, the results are not accurate. Therefore, we opted to randomly sample the indices as two-dimensional unit vectors. Our simplified concept block also follows this approach, sampling the indices randomly during training (refer to Figure 12). While CIC still struggled to effectively explore the space behind obstacles, our model ultimately showed a performance improvement of approximately 15% (refer to Table 1).

## C.5 A COMPREHENSIVE ANALYSIS OF VARIATIONS IN EXPERIMENTAL TECHNIQUES

| MI Form | Objective | Algorithm |
|---------|-----------|-----------|
| Backward | $\alpha H(z) - \beta H(z\mid s)$ | VIC (Gregor & Danilo Jimenez Rezende, 2016) |
| Backward | $-H(z\mid s) + H(a\mid z, s)$ | DIAYN (Eysenbach et al., 2019) |
| Backward | $-H(z\mid s) + H(a\mid z, s)$ | VALOR (Achiam et al., 2018) |
| Backward | $\alpha H(z) - \beta H(z\mid s)$ | VISR (Hansen et al., 2020) |
| Forward | $H(s) - H(s\mid z)$ | DADS (Sharma et al., 2020) |
| Forward | $-H(s\mid z)$ | EDL (Campos et al., 2020) |
| Forward | $H(\psi_\theta(s))$ | APT (Liu & Abbeel, 2021b) |
| Forward | $\alpha H(\psi_\theta(s)) - \beta H(\psi_\theta(s\mid z))$ | APS (Liu & Abbeel, 2021a) |
| Forward | $\alpha H(\psi_\theta(s)) - \beta H(\psi_\theta(s\mid z))$ | CIC (Laskin et al., 2022) |
| Integrated | $D_{kl}(\psi_z(s\mid z)\|\psi_z(s))$ | CB (Our Model) |

Table 2: Classification of existing algorithms based on their Mutual Information (MI) form and objective functions. 'Forward' approximates $I(S; Z) = H(Z) - H(Z\mid S)$, while 'Backward' approximates $I(S; Z) = H(S) - H(S\mid Z)$. The terms $\alpha$ and $\beta$ signify that the respective terms originate from distinct algorithms. In the proposed framework, $\psi_\theta$ is defined as the parametric embedding, and $\psi_z$ as the nonparametric embedding, which utilizes individual skills as variables.

We have carefully considered that distinguishing skills is not solely about reaching a new state by increasing entropy; instead, it sometimes requires organizing skills while reducing entropy. The most significant distinction between our method and those previously established lies in our approach to updating skills. While prior methods maintained a consistent state embedding, viewing the state from a unified perspective regardless of which skill was planned to be updated, our method introduces a novel approach by distorting the space uniquely from the perspective of each skill during updates. Our approach can be viewed as utilizing a nonparametric method that incorporates skill as a dimension. A comparison between previous algorithms and our algorithm is presented in Table 2.

Ultimately, our method allows the skills not merely to separate but to coalesce and share reusable patterns autonomously. Therefore, our method leads to reduction in the hypothesis space, providing enhanced optimization.

## D   ADDITIONAL PLOTS

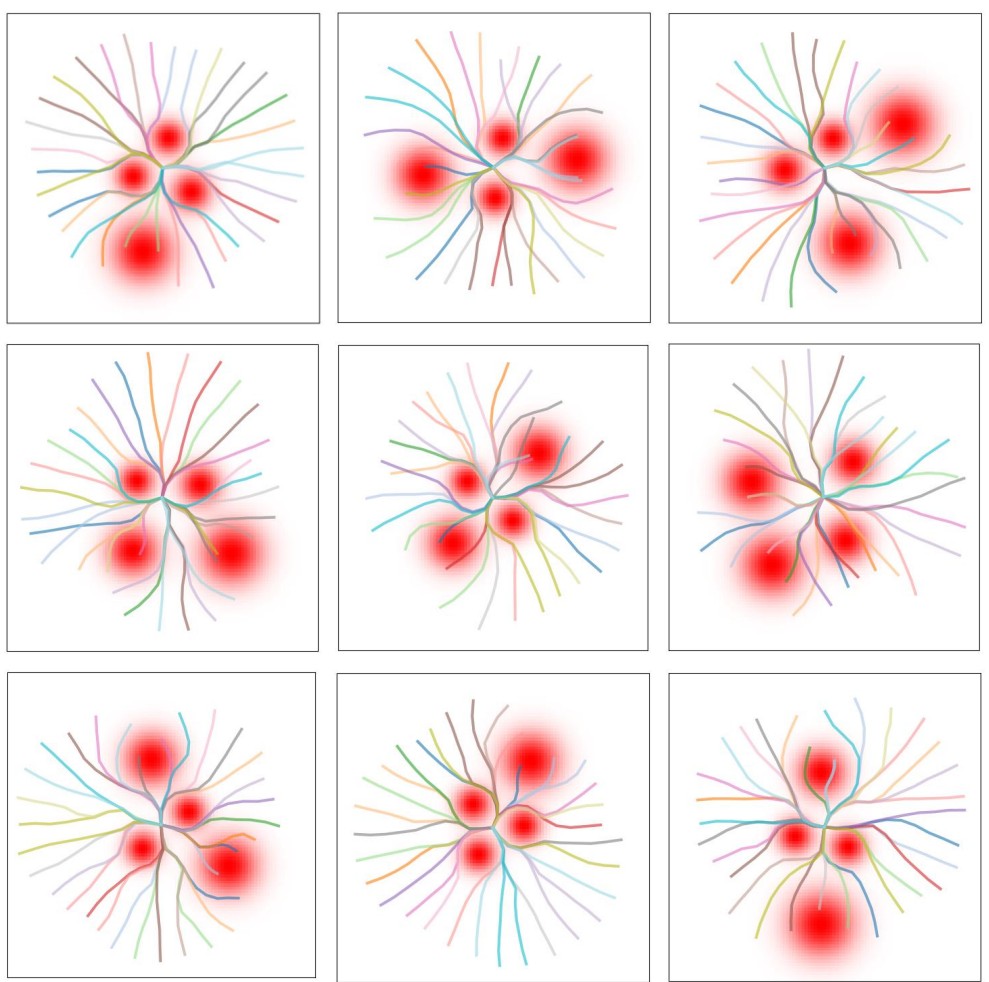

Figure 13: This graph represents the results of generating multiple maps using the method depicted in 8, and subsequently training the concept blocks accordingly.

## E    THE ULTIMATE SIGNIFICANCE OF THIS STUDY

When we engage in the learning process, we often try to fit new information into our existing frameworks of language and cognition. This approach leads us to incorporate even slightly different patterns into familiar structures, sometimes missing out on alternative perspectives that language alone cannot capture. As knowledge grows and becomes more complex, the gaps between different pieces of knowledge widen, making it harder to see how these pieces connect within our cognitive frameworks. This complexity can make it difficult to evaluate how effectively we are learning.

In response to these challenges, we break down knowledge to explore new areas, allowing for a natural optimization of knowledge through diversification. Our research demonstrates this approach, highlighting the human tendency to learn and adapt. Additionally, in the field of mathematical theorem proving, increasing problem complexity makes us more aware of the limits of our current knowledge systems. This awareness can prevent us from confirming the accuracy of our conclusions. At the edge of cognitive exploration, we narrow down the scope of our study to refine and segment established knowledge, leading to new theories and hypotheses that improve optimization and facilitate successful theorem proving. This model not only illustrates how curiosity drives knowledge optimization but also marks the beginning of an artificial intelligence capable of independently determining what is correct and what is not.

