# OpenReview forum: "CURIOSITY IS THE PATH TO OPTIMIZATION"
_ICLR.cc/2025/Conference — Submitted to ICLR 2025_

### Official Review · Reviewer_Zw9v · 2024-10-18

**Soundness:** 1
**Presentation:** 1
**Contribution:** 1
**Rating:** 1
**Confidence:** 5

**Summary:**

This paper introduces an algorithm for unsupervised skill discovery, provides some theoretical claims, and a short experiment on a 2d problem.

**Strengths:**

The title is nice, even though it is not representative of the content.

**Weaknesses:**

Everything else. It may seem rude, but the paper indeed combines a more than generous representation of what the paper indeed provides, with a very bad writing which prevents understanding the ideas the authors try to convey.

About the too generous representation of what the paper indeed provides, below are some quotes and the issue with them (the list is far from being exhaustive):
* "*we have mathematically proven that curiosity provides bounds to guarantee optimality in contrastive multi-skill RL*": this is an overclaim, neither curiosity nor the notion of optimality having been defined afterward (and it may be better to write the paper to the present than to the past, but this is quite minor)
* "_we have leveraged these findings to develop an algorithm that is applicable in real-world scenarios..._": this is very hard to assess, given all the missing details of the algorithm, but the paper provides no empirical evidence of this (sole experiment being done on a 2d environment)
* "*...which has been demonstrated to surpass other prominent algorithms*": there is hardly any empirical evidence for this
* "*we aim to discuss how curiosity itself serves as a process of reducing the hypothesis space*": there is not such discussion in the paper, and neither curiosity nor the concept of hypothesis space are formally (or even informally indeed) defined
* "*this adaptation contributes to the autonomous optimization of the hypothesis space*": the paper provides no evidence about this claim, be it theoretical or empirical (and what the sentence means is unclear btw)
* "*This research demonstrates how skills self-categorize and inherently structure themselves into tree-like formations*": there is not evidence of this, except if one consider the path figures as representing trees (but this has no generality, and the lack of details would make it impossible to reproduce)
* "*demonstrating that in our proposed model, curiosity and hypothesis space compression are identical*": this is an unsupported claim, for the reasons already provided
* Appx D, titled "*The Ultimate Significance of This Study*": it is hard to see the point of this section, its relationship to what the paper indeed proposes or provides evidences for, but it is quite representative of the overall tone of the paper.

Next are much more detailed comments about the issue with the writing, how things are defined (or not defined very often, or ill-defined), overall what makes the paper very hard to understand.

## Section 3.1, Contrastive Space
* $(S, d)$ is stated as being a metric space. Fine enough. But $d$, from the proofs, is necessarily the euclidian distance (because everything goes through Gaussians). And $S$ is $\mathbb{R}^n$ (or a subspace of it?) according to l.146. Then a projection is defined as $\psi:S\rightarrow C$, but without specifying what $C$ is (appart from say it is a "*contrastive space*"). And here a skill is implicitly defined as the result of a projection ("*two skills $\psi(s_1)$ and $\psi(s_2)$*"). All of this is already quite unclear, not formal enough, while the concepts of skill and contrastive spaces seems central to the paper
* Then we have the introduction of the function $\phi$ l.145, but does such a function exist, and if so under what conditions? Anyway, it seems to be never used afterward.
* Then we have "*$K$ is the set of skills*". But a skill is the result of a projection, so this does not seem consistant. Where does this set comes from, or is it computed, and how? Then we have "*$T$ is the set of trajectories*", but trajectories have not been defined so far (nor state, nor MDP...).
* Then $\psi_B$, which is quite central to the paper, is defined as $\psi_B(s) = (c_1, \dots, c_k)$ (and we discover that the contrastive space has dim $k$, reading between lines, but this does not tell the relationship to the set $K$). The definition l.155-157 defines $\psi_B(s)$ as a function of $T$ (the set of trajectories? what does the term $\frac{1}{T}$ mean then?), of $s_j$ and of $s_{j,t}$ (what are these two quantities wrt $s$, the argument of $\psi_B$?). One may guess that $s_j$ or $s_{i,t}$ is a state (but then what is the other term, and are they homogeneous). It also raises notations questions, is $s$ a point of the metric space, a trajectory, a state, has to be guessed from the context, something else?

So, this first technical paragraph defining a number of important concepts for what follows is already very puzzling because things are not defined clearly enough, be it mathematically (which is a strong issue given the emphasize given by the authors claiming to prove mathematically things), or even by words.

## Section 3.2, Performance Measurement in MDPs
* "*we conduct experiments across a variety of MDP environments, each characterized by a unique set of reward functions*": there seems to be a single environment considered (the 2d one), and btw none involve rewards (or at least it's unclear), and the paper is about unsupervised skill discovery, so this statement about rewards is very puzzling. More generally, why considering MOMDPs in this context? It is not justified, not explained, one don't know if the rewards are given beforehand, need to be computed, or are abstract to state some theoretical results, or something else.
* l.163, "*unlinke a standard MDP, where $R$ represents the space of possible reward functions*": this is very wrong, in a standard MDP one considers a single reward function
* l. 167, "*$s\in S$ represents an individual state*": same notation as for the metric space, so the metric space is the state space? Or just an unfortunate notation? Or something else?
* l. 171, the policy is conditioned on $z$, which has never been defined or introduced, and some noise is added to the policy (interpreting $\epsilon$ as noise, given that this also is never defined)
* l. 173, the state space being continuous, the transition kernel is a density, there is no reason for it to be bounded by $1$
* l. 185, the state distribution is modeled by a Gaussian mixture, without much justification. But the state distribution is never defined. In a $\gamma$-discounted setting, the relevant distribution should be the $\gamma$-discounted occupancy measure induced by a policy, while the definition does not involve the discount factor.
* l. 190, "*we assume that the rewards depend solely on the state, as $s_{t+1}\sim T(s_t,a_t)$*": this is a wrong justification, because you consider $R(s_t)$ afterwards, and not $R(s_{t+1})$, and both are not equivalent, the considered reward is restrictive (consider a ring MDP, you cannot define such a reward for going always right this way, as an example)
* l. 196. There are numerous problems with the equation (which is an issue, as it just defines the expected return). First, we have both $R$ and $R'$, what is the difference? $\theta$ was said to parametrize the policy, in the equation it's $\pi$ and not $\pi_\theta$, but the reward is indexed by $\theta$, this does not make sense. Btw, what the reward is, the MOMDP is defined with a set of reward functions, which one is it? Also, the equation is between the mhs (middle hand side) and the rhs is wrong. First, it would be only true if the mixture of Gaussian was perfectly estimating the state distribution, which cannot btw be true given that the discounting is just ignored.
* between the definition of $P_\theta$ and that of $\psi_B$ (as imperfect as they are), and from other places in the paper, it seems that there is an assumption that the distance between states can be usefully measured with an L2 distance, which is a very strong assumption in general, and which should be discussed.

## Section 4.1, Convex Optimization and Information Gain
Let start with the proofs of this section
### Proof of Lemma B.1 (Appx B)
The proof is very badly written, inaccurate, and sometime just wrong. Even the result of this lemma is not clearly stated (stating that when $\sigma$ goes to 0, and then writing an equation depending on $\sigma$).

One important thing ignored by the authors is that when the common variance of two Gaussians with different means goes to zero, their KL goes to infinity. Given that half of the proof consists in providing upper-bounds based on these quantities, it makes no sense. For all equations on the top of p.16, the reader doesn't even know what is attempted, it has to be guessed from what follows. Even this is really misleading, the authors write things like $$D_{KL}(\frac{X+Y}{2}||Z)$$ with $X$, $Y$ and $Z$ being gaussian random variables (of say densities $P_X$, $P_Y$ and $P_Z$). So $X+Y$ is not a mixture of Gaussian, it is a Gaussian. This is because summing random variables is not the same as summing densities. And we measure a KL between densities and not random variables. So the authors should have written $$D_{KL}(\frac{P_X+P_Y}{2}||P_Z)$$ (which corresponds to the integrals above, up to the multiple typos in the Eq).

In the end, the proof does not make sense. The final results is indeed true, without any assumption on $\sigma$ (or even on the mixture components being Gaussian), it is a very direct consequence of the KL being convex in the pair of probability measures (see wikipedia). And it does not hold only for scalar random variables, the only case considered in the proof. But this is a very basic result, and the reviewer does not intend to correct all the proofs (which would not be possible given the lack of proper definitions of the considered mathematical objects).

### Proof of Thm B.1
Why is the first inequality true, for what norm? Why is the second inequality true, due to Hölder? It also seems necessary to define the total variation. For the last line, how does one go from $L_2$ to a generic metric $d$ (because it is never said that $d$ is restricted to $L_2$)?

### Back to the section
In thm 4.1, $\hat{J}$ is defined as the estimated value of $J$, but how? Monte-carlo rollout, how many rollouts, something else? Stating a thm on an undefined object is really an issue. $R_\text{max}$ is not defined either. The terms $s_\theta,t$ also are not defined. Consequently, the result is pretty useless (one could guess, but then it should not be called a Thm).
* "*to maximize the performance between every pair of policies*": what does this mean?
* Eq (2) does not makes sense. Over what do we optimize? What are $s_i$ and $s_j$, what does $\psi_B(s)_i$ means (indexing by $i$)? This is really an issue, given that it seems to be quite important for what follows.
* All the rest is impossible to understand, even after having read Eysenbach et al paper, how does this relate to this submission, what it tells, etc.

## Sec 4.2, an intuitive examination...

Thm 4.2 is again a thm stating things about object that have not even been defined. What is $I$? what is $\tilde{\psi}$ wrt $\psi$? What is $S$? What does it mean to index by $k$ or not? Given that the statement itself of the theorem is incomprehensible, the proof has not been checked. But it was attempted (starting from Lemma B.2, in order to maybe guess what the theorem states), and it starts with an object $\psi(s_\theta)_\theta$, which is again something new and not defined.

## And so on

The rest of the paper is in the same vein, it hardly makes sense. Even the proposed algorithm and experiment are not presented with sufficient details (and the details that are there are made fuzzy by the overall poor presentation).

**Questions:**

Unfortunately, I think the issues of this paper are too numerous to be addressed during a discussion phase.

---

### Official Review · Reviewer_7W7e · 2024-11-01

**Soundness:** 3
**Presentation:** 2
**Contribution:** 2
**Rating:** 5
**Confidence:** 5

**Summary:**

This paper shows the evidence that multi-skill RL algorithm drives the optimization in the hypothesis space both theoretically and empirically.

**Strengths:**

1. The paper is easy to follow.
2. The paper provides both empirical evidence and theoretical evidence of the point that the authors try to make.

**Weaknesses:**

The experimental result is weak. Despite the proposed algorithm CB provides a seemingly more desirable trajectories compared to the baselines, the dimension of the environment is very low, hence it would be questionable whether the algorithm would actually explore the hypothesis space in a desirable way.

**Questions:**

1. What is the order of the sample complexity of the proposed algorithm CB?
2. In a more complex environment, would CB still perform well? I would love to see some results on more complex environment, at least at MiniGrid level, where the agent will have to navigate according to the image observation.
3. What is the difference between CB and APT [1] and APS [2]? They look very similar.

[1] Behavior From the Void: Unsupervised Active Pre-Training. by Hao Liu, Pieter Abbeel
[2] APS: Active Pretraining with Successor Features. by Hao Liu, Pieter Abbeel

---

### Official Review · Reviewer_5xHm · 2024-11-03

**Soundness:** 2
**Presentation:** 1
**Contribution:** 2
**Rating:** 3
**Confidence:** 4

**Summary:**

This paper asserts that curiosity can constrain the hypothesis space and offers bounds that ensure optimality in contrastive multi-skill RL, substantiated by thorough mathematical proofs. The implementation of these concepts is also demonstrated through various real-life scenarios. However, the reviewer has raised several concerns regarding the paper’s motivation, simulations, and overall presentation. These concerns are detailed in the later section.

**Strengths:**

The paper provides detailed mathematical proofs accompanied by relevant experiments, which strengthen the theoretical assertions made.

**Weaknesses:**

The abstract and introduction are challenging to comprehend. The motivation for incorporating curiosity is repeatedly emphasized without a clear explanation of its importance or an outline of the underlying principles of the approach. This discussion appears only briefly in the latter part of the paper.

The introduction lacks essential background information on the significance of the studied problem and does not introduce the PAC-MDP framework, making it difficult for readers unfamiliar with this concept to grasp the notations and follow the argumentation.

The clarity and structure of the paper require significant enhancements. The mathematical formulas are particularly difficult to follow, with numerous variables and concepts left undefined.
For instance, the variable S is not clearly defined (line 136 - 137,), and the paper claims that the policy is one-dimensional without explaining its implementation or definition. Furthermore, the representation of skill by ψ(s1) (line 138) is confusing—is s related to skill, or does it denote state, as defined later?
The variable 𝑘 (146) and parameter 𝜃 (185) are mentioned without adequate definitions.

The choice of a Gaussian distribution is employed without justification. Why is this distribution selected over others?

In Theorem 4.1, the constant C is used without clear explanation—while it's stated that C represents the contrastive space, the method by which a space is divided is not elucidated (how could you divide a space?).

These issues combine to make the paper less accessible and diminish its impact. It is crucial that these elements be addressed to improve the paper's comprehensibility and academic rigor.

**Questions:**

see Weaknesses

---

### Official Review · Reviewer_huhp · 2024-11-03

**Soundness:** 1
**Presentation:** 1
**Contribution:** 2
**Rating:** 3
**Confidence:** 3

**Summary:**

The paper studies the problem of unsupervised reinforcement learning in the contrastive space and theoretically show how the learning in the contrastive space can effective reduce the hypothesis space and reduces the sample complexity.

**Strengths:**

This paper is full of errors and inconsistent notations. It is hard to understand and evaluate the contribution without a clear presentation of the major results, especially this paper has a theoretical focus.

**Weaknesses:**

1. This paper looks like a information geometry analysis of unsupervised RL in the contrastive space. However, the equations seem to be directly copied from multiple different papers and they are mostly inconsistent and have undefined symbols. I list a few as follows:

* $T$ is referred to as a set of trajectories (line 148), transition function (line 173), and time steps (line 186).
* Information gain $I$ first appears in Theorem 4.2, but it was never formally defined. Also, the notation $I$ was already used to refer to identity matrix (line 189).
* $C$ is defined as the contrastive space (line 138) and also a small variance (line 188).
* In eq. 1, $\theta$ first appears as a subscript of states $s$, but it was not defined.
* $\tilde{\psi}$ first appears in Theorem 4.2, but it was never defined. Also, $\psi$ was defined as a mapping from state space $S$ to contrastive space $C$, but in Theorem 4.2, $\psi(S)$ takes as input the state space. Not sure what this means.
* In line 321, $ d/dx I(\psi_B(S)_\theta||\theta)$ takes derivative to $x$. Here $x$ is not defined also not sure what the subscript $\theta$ means here.
* Line 353 starts to use $s$ to denote skills, but $s$ is defined as states.

2. Theorem 4.1 does not help the discussion, not sure the purpose of introducing Theorem 4.1.

3. Eq. (2) is described as maximizing the minimum distance between any two skills, but the objective actually maximizes the sum of the distances to the mean of all the skills in the contrastive space.

4. The paper assumes all the state visitation is a "point distribution" as well as their mapping to the contrastive space. In this case, the KL divergence is illy-defined as they goes to infinity for any two different states. I don't know how the information gain is computed.

**Questions:**

Please see my weakness points.

---

### Meta-Review · Area_Chair_bBNp · 2024-12-20

**Metareview:**

This paper introduces an algorithm for unsupervised skill discovery and provides some theoretical claims about how the learning in the contrastive space can effective reduce the hypothesis space and reduces the sample complexity. The paper includes a short experiment on a 2d problem. Reviewers appreciated that the paper included both mathematical proofs accompanied and relevant experiments. However, reviewers had concerns about the clarity of the theoretical results (e.g., undefined symbols), which made it challenging to assess the correctness and precise scope of the contributions of the paper (see, especially, comments from Reviewer Zw9v). Expanding the introduction to include more context for the problem and its relationship to prior work may make the paper easier for readers to understand. Reviewers also pointed to several symbols and equations that were undefined; clarifying these is important in future versions of the paper. These concerns, which went unaddressed during the rebuttal, compel me to recommend rejecting this paper.

**Additional Comments On Reviewer Discussion:**

The authors did not post a rebuttal, and there was no subsequent discussion as the reviewers seemed nearly unanimous (scores = 3/3/3/6).

---

### Decision · Program_Chairs · 2025-01-22

Reject